# BoRA: Towards More Expressive Low-Rank Adaptation with Block Diversity

**Shiwei Li**[1,2], **Xiandi Luo**[1], **Haozhao Wang**[1,*], **Xing Tang**[2], **Ziqiang Cui**[3], **Dugang Liu**[4]
**Yuhua Li**[1], **Yichen Li**[1], **Xiuqiang He**[2,*], **Ruixuan Li**[1,*]

[1]Huazhong University of Science and Technology [2]Shenzhen Technology University
[3]City University of Hong Kong [4]Shenzhen University

`{lishiwei,hz_wang,rxli}@hust.edu.cn, hexiuqiang@sztu.edu.cn`

## Abstract

Low-rank adaptation (LoRA) is a parameter-efficient fine-tuning (PEFT) method widely used in large language models (LLMs). It approximates the update of a pretrained weight matrix $W \in \mathbb{R}^{m \times n}$ by the product of two low-rank matrices, $BA$, where $A \in \mathbb{R}^{r \times n}$ and $B \in \mathbb{R}^{m \times r}$ ($r \ll \min\{m, n\}$). Increasing the dimension $r$ can raise the rank of LoRA weights (i.e., $BA$), which typically improves fine-tuning performance but also significantly increases the number of trainable parameters. In this paper, we propose **Block Diversified Low-Rank Adaptation (BoRA)**, which improves the rank of LoRA weights with a small number of additional parameters. Specifically, BoRA treats the product $BA$ as a block matrix multiplication, where $A$ and $B$ are partitioned into $b$ blocks along the columns and rows, respectively (i.e., $A = [A_1, \ldots, A_b]$ and $B = [B_1, \ldots, B_b]^\top$). Consequently, the product $BA$ becomes the concatenation of the block products $B_i A_j$ for $i, j \in [b]$. To enhance the diversity of different block products, BoRA introduces a unique diagonal matrix $\Sigma_{i,j} \in \mathbb{R}^{r \times r}$ for each block multiplication, resulting in $B_i \Sigma_{i,j} A_j$. By leveraging these block-wise diagonal matrices, BoRA increases the rank of LoRA weights by a factor of $b$ while only requiring $b^2 r$ additional parameters. Extensive experiments across multiple datasets and models demonstrate the superiority of BoRA, and ablation studies further validate its scalability. The code is available at `https://github.com/Leopold1423/bora-iclr26`.

## 1 Introduction

Large language models (LLMs) have demonstrated remarkable performance across a wide range of tasks (DeepSeek-AI, 2024; OpenAI, 2024). However, the state-of-the-art models usually contain a vast number of parameters, making full fine-tuning (FFT) for downstream tasks extremely expensive in terms of both training time and memory usage, thereby limiting their practical deployment (Hu et al., 2022; Lester et al., 2021). To mitigate this issue, parameter-efficient fine-tuning (PEFT) methods have been developed to reduce the number of trainable parameters and decrease fine-tuning costs (Xu et al., 2023). These methods include techniques such as prompt tuning (Lester et al., 2021), parallel adapters (He et al., 2022) and sequential adapters (Houlsby et al., 2019). Among these, low-rank adaptation (LoRA) (Hu et al., 2022) has gained popularity due to its ability to avoid additional inference latency. Specifically, LoRA freezes the pretrained weight matrix $W \in \mathbb{R}^{m \times n}$ and learns two smaller low-rank matrices to approximate the weight update as $\Delta W = \alpha/r BA$, where $A \in \mathbb{R}^{r \times n}$, $B \in \mathbb{R}^{m \times r}$, $r$ is the LoRA rank ($r \ll \min\{m, n\}$), and $\alpha$ is an adjustable scaling factor. For simplicity, we omit the coefficient $\alpha/r$ in the subsequent description.

Despite its effectiveness, LoRA generally exhibits a performance gap compared to FFT, which is typically attributed to the limited rank of LoRA weights (Hu et al., 2022; Ren et al., 2024; Huang et al., 2025). Specifically, the rank of LoRA weights is constrained as $\text{rank}(\Delta W) \leq \min\{\text{rank}(A), \text{rank}(B)\} \leq r$. Numerous studies have also demonstrated that increasing the rank of LoRA weights typically improves fine-tuning performance (Ren et al., 2024; Liu et al., 2024; Huang et al., 2025). This trend is further corroborated by our experimental results, as shown in Tables 1, 2, and 3. Recently, Zeng and Lee (2024) further explored LoRA's expressive power, using the LoRA rank $r$ to quantify the approximation error between the LoRA weights (i.e., $\Delta W = BA$) and the



Figure 1: An illustration of BoRA in comparison to LoRA (Hu et al., 2022) and MELoRA (Ren et al., 2024). (a) **LoRA** weights (i.e., $BA$) can be represented by block matrix multiplication, where $A = [A_1, A_2, A_3]$ and $B = [B_1, B_2, B_3]^\top$. (b) **MELoRA** zeros out the off-diagonal blocks to break the correlation between different blocks and enhance the rank of LoRA weights. (c) **BoRA** introduces a diagonal matrix for each block multiplication to enhance the diversity among block products. Notably, LoRA and MELoRA are essentially specific instances of BoRA. BoRA reduces to LoRA when all $\Sigma_{i,j} = I$, and to MELoRA when $\Sigma_{i,j} = I$ for $i = j$ and $\Sigma_{i,j} = 0$ for $i \neq j$, where $I$ denotes the identity matrix.

assumed optimal weight update. Their findings suggest that a higher rank of LoRA weights leads to a smaller approximation error. Consequently, efficiently increasing the rank of LoRA weights has become a consensus strategy for improving fine-tuning performance.

In this paper, we analyze the rank of LoRA weights through the lens of block matrix multiplication. Let matrices $A$ and $B$ be evenly partitioned into three blocks along the columns and rows, respectively, i.e., $A = [A_1, A_2, A_3]$ and $B = [B_1, B_2, B_3]^\top$. The product $BA$ can then be represented as the concatenation of the block products $B_i A_j$, as shown in Figure 1(a). This structure limits the independence of different block products, thereby constraining the rank of $BA$. Specifically, the rank of $BA$ depends on the number of linearly independent row vectors; however, the difference between block rows lies solely in the use of different $B_i$. This implies that the blocks in one row can be obtained from those in another row using the same transformation. For instance, left-multiplying the blocks of the first row by $B_2 B_1^{-1}$ (assuming $B_1^{-1}$ exists) generates the second row, indicating that the second row does not contribute to the rank. A similar issue arises with the columns.

To address this issue, we propose **Block-Diversified Low-Rank Adaptation (BoRA)**, which effectively breaks the correlation between the block products $B_i A_j$ in different rows or columns. As illustrated in Figure 1(c), BoRA introduces a unique diagonal matrix, $\Sigma_{i,j}$, for each block multiplication, resulting in $B_i \Sigma_{i,j} A_j$. Assuming matrices $A$ and $B$ are divided into $b$ blocks, BoRA will increase the rank of LoRA weights by a factor of $b$, while requiring only $b^2 r$ additional parameters. In contrast, MELoRA (Ren et al., 2024) achieves a similar effect by disrupting the correlation of different blocks through zeroing out the off-diagonal block products, as illustrated in Figure 1(b). Although this increases the rank of LoRA weights, it may limit LoRA's expressive power due to the presence of numerous zero entries. Instead, BoRA ensures block diversity by utilizing block-wise diagonal matrices, without compromising LoRA's expressiveness. Importantly, both LoRA and MELoRA can be seen as specific instances of BoRA. Specifically, BoRA reduces to LoRA when all $\Sigma_{i,j} = I$, and to MELoRA when $\Sigma_{i,j} = I$ for $i = j$ and $\Sigma_{i,j} = 0$ for $i \neq j$, where $I$ denotes the identity matrix.

**Our contributions can be summarized as follows:**

- We analyze LoRA from the perspective of block matrix multiplication, revealing that the rank of LoRA weights is constrained due to correlations between different block products.

- We propose BoRA, which break the correlation among different block products with block-wise diagonal matrices. By dividing matrices $A$ and $B$ into $b$ blocks, BoRA increases the rank of LoRA weights by a factor of $b$, requiring only $b^2 r$ additional parameters.

- We conduct extensive experiments on multiple models and datasets, demonstrating that BoRA consistently outperforms LoRA and its variants. Using a similar number of trainable parameters, BoRA can achieve 2-4% accuracy improvement over LoRA.

## 2 RELATED WORK

To reduce fine-tuning overhead, LoRA (Hu et al., 2022) decomposes the weight update, $\Delta W \in \mathbb{R}^{m \times n}$, into two low-rank matrices, $A \in \mathbb{R}^{r \times n}$ and $B \in \mathbb{R}^{m \times r}$, where the LoRA rank $r$ determines the number of trainable parameters. LoRA can be integrated into a model without altering its architecture or increasing inference overhead. In contrast, DoRA (Liu et al., 2024) decomposes pretrained weights into direction and magnitude components, learning the magnitude via a trainable vector and updating the direction with LoRA. Recent studies have explored various aspects of LoRA to improve its performance, such as the scaling factor (Kalajdzievski, 2023), initialization methods (Meng et al., 2024; Wang et al., 2024), learning rates (Hayou et al., 2024), and dynamic parameter allocation (Zhang et al., 2023; Li et al., 2024). For example, rsLoRA (Kalajdzievski, 2023) modifies the scaling factor to $\alpha/\sqrt{r}$ for more stable fine-tuning. PiSSA (Meng et al., 2024) and LoRA-GA (Wang et al., 2024) conduct singular value decomposition (SVD) on pretrained weights and sampled gradients to initialize the matrices $A$ and $B$ of LoRA. LoRA+ (Hayou et al., 2024) suggests that using a higher learning rate for matrix $B$ can improve fine-tuning performance. AdaLoRA (Zhang et al., 2023) adaptively adjusts the LoRA rank for different layers during fine-tuning, allocating more parameters to important layers within a fixed parameter budget. VB-LoRA (Li et al., 2024) composites all the low-rank matrices of different LoRA layers from a shared vector bank.

In this paper, we focus on LoRA variants that efficiently enhance the rank of LoRA weights. For example, HiRA (Huang et al., 2025) and KronA (Edalati et al., 2023) use the Hadamard and Kronecker products, respectively, to improve the rank of LoRA weights. ReLoRA (Lialin et al., 2024) periodically merges learned LoRA adapters into the pretrained weights to increase the rank of weight updates. MELoRA (Ren et al., 2024) achieves a higher rank by stacking low-rank matrices along the diagonal. However, MELoRA introduces many zero values, which can significantly reduce the expressiveness of LoRA, as shown in Figure 1(b). In contrast to previous approaches, we analyze the rank of LoRA weights through the lens of block matrix multiplication. Our proposed BoRA increases the diversity of block products by introducing unique diagonal vectors for each block multiplication. Notably, both LoRA and MELoRA are special cases of BoRA, as illustrated in Figure 1. Furthermore, MoELoRA (Luo et al., 2024) trains multiple LoRA adapters as distinct experts and combines their knowledge via a routing network. HydraLoRA (Tian et al., 2024) improves on this by sharing the matrix $A$ across the MoELoRA framework for more efficient adaptation. Essentially, MoELoRA can also be viewed as a form of block matrix multiplication. The main difference between BoRA and MoELoRA lies in the way matrices $A$ and $B$ are partitioned, which will be discussed in detail in Section 3.5. Additional discussions of other PEFT-related methods are deferred to Appendix E.

## 3 METHODOLOGY

### 3.1 BLOCK MATRIX MULTIPLICATION

Block matrix multiplication is a fundamental operation that partitions matrices into smaller sub-matrices for efficient multiplication. Let $M \in \mathbb{R}^{m \times r}$ and $N \in \mathbb{R}^{r \times n}$, where $M$ and $N$ are evenly divided into $b_m \times b_r$ and $b_r \times b_n$ sub-matrices, respectively. Then, the product $P = MN$ can be computed block by block, with each block $P_{i,j}$ ($i \in [b_m], j \in [b_n]$) calculated as follows:

$$P_{i,j} = \sum_{k=1}^{b_r} M_{i,k} N_{k,j}. \tag{1}$$

In this paper, the matrices $A \in \mathbb{R}^{r \times n}$ and $B \in \mathbb{R}^{m \times r}$ of LoRA are evenly partitioned into $b$ blocks along the columns and rows, respectively (i.e., $A = [A_1, \ldots, A_b]$ and $B = [B_1, \ldots, B_b]^\top$). Consequently, each block of the LoRA weights $\Delta W_{i,j}$ ($i, j \in [b]$) can be expressed as:

$$\Delta W_{i,j} = B_i A_j. \tag{2}$$

Previous studies have shown that the rank of LoRA weights significantly affects the fine-tuning performance (Ren et al., 2024; Huang et al., 2025; Lialin et al., 2024). However, the rank of LoRA

weights is generally limited by the dimension $r$ (i.e., the LoRA rank), regardless of the dimensions $m$ and $n$, as follows:

$$\text{rank}(\Delta W) = \text{rank}(BA) \leq \min\{\text{rank}(A), \text{rank}(B)\} \leq r. \tag{3}$$

From the perspective of block matrix multiplication, this issue primarily stems from the lack of independence between block products $(B_i A_j)$ across different rows or columns. As shown in Figure 1(a), the block products share the same $B_i$ for each row and the same $A_j$ for each column. Taking rows as an example, the rank of $\Delta W$ depends on the number of linearly independent row vectors. However, the difference between block rows lies only in the use of different $B_i$. This implies that blocks in one row can be derived from blocks in another row using the same transformation. For example, in Figure 1(a), if $\text{rank}(B_1) = r$, then $B_1^{-1}$ exists. Left-multiplying the first row block by $B_2 B_1^{-1}$ generates the second row, indicating that the second row does not contribute to the rank. A similar issue applies to the columns. Breaking the correlation between these blocks can increase the rank of the weight matrix, thereby improving the expressiveness of LoRA.

## 3.2 BLOCK-DIVERSIFIED LOW-RANK ADAPTATION

To break the correlation between different block products, we propose Block-Diversified Low-Rank Adaptation (BoRA). Specifically, BoRA introduces a unique diagonal matrix for each block multiplication to enhance block diversity, as shown in Figure 1(c). Assuming $A$ and $B$ are divided into $b$ blocks along columns and rows, BoRA will additionally learn a set of diagonal matrices $\{\Sigma_{i,j} \in \mathbb{R}^{r \times r} \mid i, j \in [b]\}$, such that the multiplication of each block pair is computed as follows:

$$\Delta W_{i,j} = B_i \Sigma_{i,j} A_j. \tag{4}$$

These block-diagonal matrices, $\Sigma_{i,j}$, amplify the differences in block products across rows or columns, thus enhancing the expressiveness of LoRA. Therefore, the core concept of BoRA lies in learning block-wise diagonal matrices $\{\Sigma_{i,j} \in \mathbb{R}^{r \times r} \mid i, j \in [b]\}$, where the corresponding parameters are represented by a three-dimensional tensor $\sigma \in \mathbb{R}^{b \times b \times r}$. To ensure $\sigma$ can be effectively optimized with the same learning rate as the matrices $A$ and $B$, we initialize $\sigma$ using the same Kaiming initialization (He et al., 2016) applied to the matrix $A$. Furthermore, to facilitate the learning of $\Sigma$, we normalize $\sigma$ by its mean absolute value and apply the exponential function to generate $\Sigma$ as follows:

$$\Sigma_{i,j} = \text{Diag}(\text{Exp}(\frac{\sigma[i][j]}{\text{Mav}(\sigma)})), \tag{5}$$

where $\text{Mav}(\sigma) = \frac{\|\sigma\|_1}{b^2 r}$ denotes the mean absolute value of $\sigma$, $\text{Exp}(\cdot)$ denotes the exponential function, and $\text{Diag}(\cdot)$ denotes the diagonalization function, which converts a vector into a diagonal matrix. Since $\sigma[i][j]$ is initialized with a small variance, the differences between $\sigma[i][j]$ across blocks are minimal, which limits the diversity between blocks. Normalizing by the mean absolute value of $\sigma$ can reduce the impact of its small initialization value on the distribution and optimization of $\Sigma$. Additionally, because $\text{Norm}(\sigma[i][j]) = \frac{\sigma[i][j]}{\text{Mav}(\sigma)}$ is still zero-centered, zero or near-zero values could nullify the values in $B_i$ and $A_j$. To prevent information loss associated with a zero value in $\Sigma$, we further apply the exponential function to $\sigma$, ensuring that $\Sigma$ contains only positive values and preventing any loss of information due to zero or near-zero values.

## 3.3 RANK UPPER BOUND OF BoRA

In the previous discussion, we presented the motivation and formulation of BoRA by expressing its weight as the concatenation of block products $(B_i A_j)$. In fact, the weight in BoRA can also be represented as the product of three matrices, as shown below:

$$\Delta W = B' \Sigma' A', \tag{6}$$

where $A' \in \mathbb{R}^{br \times n}$ and $B' \in \mathbb{R}^{m \times br}$ are diagonal block matrices formed from the blocks $\{A_1, \ldots, A_b\}$ and $\{B_1, \ldots, B_b\}$, respectively. $\Sigma' \in \mathbb{R}^{br \times br}$ is the matrix obtained by concatenating all $\Sigma_{ij}$ for $i, j \in [b]$. It is clear that the rank of each of these three matrices is bounded above by $br$. Using the properties of rank in matrix multiplication (i.e., $\text{rank}(BA) \leq \min\{\text{rank}(A), \text{rank}(B)\}$), we can derive an upper bound for the rank of the weights in BoRA, as stated in Proposition 1.

**Proposition 1** (The Rank Upper Bound of BoRA). *Using the low-rank matrices $A = [A_1, \ldots, A_b] \in \mathbb{R}^{r \times n}$ and $B = [B_1, \ldots, B_b]^\top \in \mathbb{R}^{m \times r}$, along with a set of diagonal matrices $\{\Sigma_{i,j} \in \mathbb{R}^{r \times r} \mid i, j \in [b]\}$, the weight update generated by BoRA, denoted as $\Delta W$, satisfies*

$$rank(\Delta W) \leq \min\{m, n, br\}, \tag{7}$$

*where the rank upper bound can be achieved when all three matrices are full rank (i.e., $rank(B') = \min\{m, br\}$, $rank(A') = \min\{n, br\}$, and $rank(\Sigma') = br$).*

The detailed theoretical justification for Proposition 1 is provided in Appendix F. Note that, by using the same matrices $A$ and $B$, the rank of the LoRA weights is constrained by $r$. Proposition 1 demonstrates that BoRA requires only $b^2 r$ additional parameters to increase the weight rank by a factor of $b$. In contrast, LoRA needs $(m + n)br$ parameters to achieve the same bound.

On the other hand, to achieve a target rank $R = br$, the number of parameters required by BoRA is $N = (m + n + b^2)r = R(\frac{m+n}{b} + b)$, and this quantity is minimized when $b = \sqrt{m + n}$. From the perspective of parameter efficiency, this suggests that setting $b \approx \sqrt{m + n}$ is a good practical choice. This optimal value arises from a natural trade-off: the achievable rank of BoRA (i.e., $br$) increases linearly with $b$, while the additional parameter cost (i.e., $b^2 r$) grows quadratically. Once $b$ exceeds $\sqrt{m + n}$, further increasing $b$ to obtain a higher rank becomes less efficient than simply increasing the base rank $r$ directly. Therefore, in practice, one can simply set $b = \lfloor \sqrt{n} \rfloor$ for BoRA.

## 3.4 EFFICIENT FORWARD PROPAGATION OF BoRA

In this section, we discuss the forward propagation process of BoRA. Assuming the input token is $X \in \mathbb{R}^n$, the forward propagation of LoRA is defined as $Y = WX + BAX$, where $BAX$ represents the LoRA output. Differently, BoRA divides the input token into several segments to efficiently perform block matrix multiplication. As shown in Figure 2, if the matrices $A$ and $B$ are partitioned into $b$ blocks, BoRA also evenly divides the input token $X$ into $b$ segments. These segments, denoted as $\{X_1, \ldots, X_b\}$, are processed to produce $b$ output segments, $\{Y_1, \ldots, Y_b\}$, which are then concatenated to form the final BoRA output. Each output segment $Y_j$ is of the following form:

$$Y_j = B_j \sum_{k=1}^{b} \Sigma_{j,k} A_k X_k. \tag{8}$$

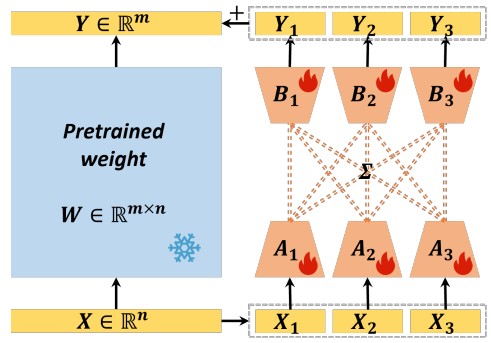

Figure 2: BoRA divides the input token $X$ into several segments to efficiently perform block matrix multiplication. The dotted line connecting $A_j$ and $B_i$ represents the trainable diagonal matrix $\Sigma_{i,j}$.

Notably, compared to LoRA, BoRA only requires the additional multiplication of the diagonal matrix $\Sigma_{j,k} \in \mathbb{R}^{r \times r}$ with the vector $A_k X_k \in \mathbb{R}^r$. Since $\Sigma_{j,k}$ is a diagonal matrix, this operation can be efficiently performed through element-wise multiplication. Formally, the floating-point operations (FLOPs) per token in the forward propagation of LoRA is given by $mn + (m + n)r$, where $mn$ and $(m + n)r$ represent the computational overheads of the pretrained model and the LoRA module, respectively. Taking this additional overhead into account, the FLOPs per token in the forward propagation of BoRA is $mn + (m + n)r + b^2 r$, where $b^2 r$ represents the extra computation introduced by the block-wise diagonal matrix. Notably, the values of $mn + (m + n)r$ and $mn + (m + n)r + b^2 r$ are also the number of trainable parameters in LoRA and BoRA, respectively. This indicates that the computational density of BoRA is equivalent to that of LoRA. It is important to note that, since $b \ll \min\{m, n\}$, the additional memory and computational costs of BoRA is typically negligible.

## 3.5 COMPARISON WITH LoRA AND MELoRA

In this section, we analyze the relationships between BoRA, LoRA (Hu et al., 2022), and MELoRA (Ren et al., 2024). As shown in Figure 1, all three methods can be represented as block matrix multiplications.

LoRA is actually a special case of BoRA, where all $\Sigma_{i,j}$ matrices are set to the identity matrix $I$. In contrast, the $\Sigma_{i,j}$ matrices in BoRA follow different distributions across various $i$ and $j$, introducing diversity among the block products and enhancing the expressiveness of BoRA compared to LoRA. MELoRA enhances LoRA by setting the off-diagonal blocks to zero, thereby breaking correlations between different blocks and increasing the rank of the LoRA weights. Similarly, MELoRA is also a special case of BoRA, where $\Sigma_{i,j} = I$ for $i = j$ and $\Sigma_{i,j} = 0$ for $i \neq j$. In contrast to MELoRA, which zeros out the off-diagonal blocks, BoRA employs block-diagonal matrices to increase block diversity, thereby improving the rank of LoRA weights without introducing any zero values.

On the other hand, MoELoRA (Luo et al., 2024) is another form of block matrix multiplication for LoRA. In this paper, we partition the matrices $A = [A_1, \ldots, A_b]$ and $B = [B_1, \ldots, B_b]^\top$ into blocks along columns and rows, respectively. Conversely, if $A = [A_1, \ldots, A_b]^\top$ and $B = [B_1, \ldots, B_b]$ are partitioned along the rows and columns, we can derive the LoRA weights as follows:

$$\Delta W = \sum_{k=1}^{b} B_k A_k, \tag{9}$$

where each $B_k A_k$ can be viewed as an expert. Further, MoELoRA integrates LoRA with the Mixture-of-Experts architecture, using a routing network to adaptively combine the knowledge of different experts ($B_k A_k$). In the next section, we compare BoRA's performance with these methods in detail.

## 4 EXPERIMENTS

In this section, we evaluate BoRA on three benchmarks using various model architectures, and then perform ablation studies to assess its scalability and visualize its singular values.

### 4.1 EXPERIMENTAL SETTINGS

**Models and Datasets.**   First, we assess BoRA's natural language understanding (NLU) capability on the GLUE benchmark (Wang et al., 2019), using RoBERTa-Base (Liu et al., 2019) and RoBERTa-Large (Liu et al., 2019). The GLUE benchmark consists of eight sub-tasks, each with its own training and test sets. For each sub-task, we fine-tune the model on the training set and evaluate its accuracy on the corresponding test set. Next, we evaluate BoRA's natural language generation (NLG) capabilities on mathematical reasoning (Math10K) (Hu et al., 2023) and commonsense reasoning (Commonsense170K) benchmarks (Hu et al., 2023). Both benchmarks include a training corpus and multiple test sub-tasks. For each benchmark, we fine-tune the models on the training data and then assess their performance across all sub-tasks. To demonstrate the versatility of BoRA, we perform experiments across various model architectures and scales, including Gemma-7B (Mesnard et al., 2024), LLaMA-3-8B (Dubey et al., 2024), and Qwen2.5-14B (Yang et al., 2024).

**Baseline Methods.**   **BoRA** is compared with several baseline methods to demonstrate its effectiveness, including **LoRA** (Hu et al., 2022), along with several LoRA variants: **DoRA** (Liu et al., 2024), **MELoRA** (Ren et al., 2024) and **HydraLoRA** (Tian et al., 2024). DoRA decomposes pretrained weights into magnitude and direction components. It learns the magnitude by training a learnable vector and uses LoRA to update the direction, enhancing learning capacity while maintaining parameter efficiency. MELoRA introduces mini-ensemble low-rank adapters that collectively achieve high-rank expressive power while requiring far fewer trainable parameters than standard LoRA. HydraLoRA improves upon LoRA by employing an asymmetric architecture that increases parameter efficiency. This architecture uses one $A$ matrix and multiple $B$ matrices, which are combined through a router. These three methods represent distinct approaches to improving LoRA: better optimization, higher rank, and a mixture-of-experts architecture. A comparison with these methods clearly highlights the superiority of BoRA.

**Implementation Details.**   All experiments were conducted using NVIDIA H20 GPUs. The general settings included the AdamW optimizer (Loshchilov and Hutter, 2019), linear learning rate decay, a LoRA dropout rate of 0.05, and no weight decay. For the GLUE benchmark, we applied a warm-up ratio of 0.03. The learning rates for RoBERTa-Base and RoBERTa-Large were set to 3e-4 and 1e-4, respectively. Additionally, the number of training epochs varies across different sub-tasks; further

details are provided in Appendix B. By default, BoRA and the baseline methods were applied to the query and value weights. For mathematical and commonsense reasoning tasks, we employed a learning rate of 1e-4 with 100 warm-up steps and trained for one epoch. In these tasks, BoRA and baseline methods were applied to the query, key, and value weights. Our implementation builds upon the code from (Hu et al., 2023). In all experiments, we evaluate LoRA's accuracy across ranks 8, 16, and 32 to analyze the trade-off between parameter count and model performance. For BoRA, we set the LoRA rank to 8 and test with 8 and 16 blocks, respectively. For the three LoRA variants, we adjust the number of trainable parameters to align with LoRA at rank 8. Specifically, the rank of DoRA was set to 8, while the rank of HydraLoRA was set to 4 with three $B$ matrices. For MELoRA, the rank of mini LoRAs was set to 8, with four mini LoRA groups. Each experiment was repeated three times, and the average results were reported. Further details, including standard deviations, are provided in Appendices B and C.

## 4.2 OVERALL PERFORMANCE

**Results on NLU Tasks.** As shown in Table 1, BoRA consistently outperforms other methods on the GLUE benchmark. Specifically, BoRA achieves a 2% improvement in average accuracy over LoRA at the same rank ($r = 8$). It is worth noting that the additional parameters introduced by BoRA with 8 or 16 blocks are minimal compared to the original LoRA parameters. Even when the rank of LoRA is increased to 32, BoRA still maintains a comparable or even superior average accuracy. Regarding other LoRA variants, while some show modest accuracy improvements, they still lag significantly behind BoRA. Using a similar number of parameters, BoRA outperforms the best baseline by up to 1% on RoBERTa-Base and 2% on RoBERTa-Large. Moreover, doubling the number of blocks enhances BoRA's performance. The impact of block numbers will be examined in Section 4.4, where it is shown that further increasing the number of blocks continues to yield performance gains.

Table 1: The accuracy on **General Language Understanding** tasks with various pretrained models.

| | | #Params | RTE | MRPC | STS-B | CoLA | SST-2 | QNLI | MNLI | QQP | Avg |
|---|---|---|---|---|---|---|---|---|---|---|---|
| RoBERTa-Base | LoRA($r = 8$) | 0.29M | 72.56 | 87.25 | 87.12 | 56.10 | 93.46 | 91.58 | 84.89 | 87.46 | 82.55 |
| | LoRA($r = 16$) | 0.59M | 72.92 | 87.99 | 87.46 | 55.21 | 93.81 | 91.89 | 85.52 | 87.79 | 82.82 |
| | LoRA($r = 32$) | 1.18M | 75.09 | 89.22 | 88.01 | 58.58 | 93.58 | 90.12 | 85.84 | 88.37 | 83.60 |
| | DoRA($r = 8$) | 0.31M | 72.92 | 87.75 | 87.58 | 55.14 | 93.12 | 91.26 | 85.02 | 87.41 | 82.53 |
| | MELoRA($r = 8$) | 0.29M | 71.48 | 87.50 | 87.54 | 52.12 | 92.78 | 90.96 | 84.68 | 86.83 | 81.74 |
| | HydraLoRA($r = 4$) | 0.35M | 71.84 | 89.46 | 88.07 | 55.32 | 93.81 | 91.62 | 85.27 | 87.25 | 82.83 |
| | **BoRA**($r = 8, b = 8$) | 0.31M | 75.45 | 88.73 | 88.17 | 58.12 | 93.92 | 91.82 | 85.09 | 87.93 | 83.65 |
| | **BoRA**($r = 8, b = 16$) | 0.34M | 76.90 | 88.24 | 89.31 | 57.35 | 93.12 | 91.78 | 85.69 | 87.86 | **83.78** |
| RoBERTa-Large | LoRA($r = 8$) | 0.79M | 71.96 | 88.40 | 89.88 | 59.76 | 95.41 | 93.07 | 88.67 | 87.89 | 84.38 |
| | LoRA($r = 16$) | 1.57M | 77.74 | 88.48 | 90.60 | 61.23 | 95.57 | 93.72 | 89.29 | 88.25 | 85.61 |
| | LoRA($r = 32$) | 3.15M | 81.35 | 89.64 | 91.45 | 60.90 | 95.60 | 93.76 | 89.52 | 88.61 | 86.35 |
| | DoRA($r = 8$) | 0.84M | 75.21 | 87.83 | 89.94 | 59.47 | 95.41 | 92.99 | 88.58 | 87.84 | 84.66 |
| | MELoRA($r = 8$) | 0.79M | 72.36 | 87.32 | 86.85 | 59.37 | 95.30 | 92.70 | 88.37 | 87.33 | 83.70 |
| | HydraLoRA($r = 4$) | 0.93M | 74.01 | 89.22 | 89.47 | 59.90 | 95.41 | 92.88 | 88.87 | 87.98 | 84.72 |
| | **BoRA**($r = 8, b = 8$) | 0.81M | 78.34 | 89.95 | 91.18 | 60.34 | 95.18 | 93.48 | 89.66 | 88.52 | 85.83 |
| | **BoRA**($r = 8, b = 16$) | 0.88M | 83.75 | 88.97 | 91.33 | 60.57 | 95.30 | 93.76 | 89.81 | 88.60 | **86.51** |

**Results on Mathematical Reasoning Tasks.** As shown in Table 2, BoRA demonstrates significant improvements in mathematical reasoning tasks. At the same rank ($r = 8$), BoRA's accuracy across the three models is, on average, 2.4% higher than that of LoRA. Notably, even when LoRA's rank is increased by four times, the average improvement is only 1.9%. This highlights BoRA's superiority in increasing matrix rank and its enhanced expressiveness compared to LoRA. The improvements from DoRA and HydraLoRA are similarly modest. Although MELoRA also increases the matrix rank, it introduces many zero elements, resulting in information loss and thus diminishing performance.

**Results on Commonsense Reasoning Tasks.** As shown in Table 3, BoRA also achieves the best accuracy in commonsense reasoning tasks. The experimental conclusions are highly consistent with those in mathematical reasoning tasks. Note that the accuracy of commonsense reasoning tasks is generally higher than that of mathematical reasoning tasks, and the relative improvement is not so obvious. Nevertheless, at the same rank ($r = 8$), BoRA's accuracy across the three models is, on average, 0.95% higher than that of LoRA. In comparison, the average improvement from a fourfold

Table 2: The accuracy on **Mathematical Reasoning** tasks with various pretrained models.

| | | #Params | AddSub | MultiArith | SingleEq | GSM8K | AQuA | SVAMP | Avg |
|---|---|---|---|---|---|---|---|---|---|
| Gemma-7B | LoRA($r = 8$) | 4.82M | 87.59 | 90.33 | 89.76 | 56.10 | 29.13 | 75.70 | 71.44 |
| | LoRA($r = 16$) | 9.63M | 86.84 | 92.83 | 89.57 | 58.15 | 30.71 | 74.90 | 72.17 |
| | LoRA($r = 32$) | 19.3M | 86.58 | 91.50 | 91.93 | 58.45 | 32.28 | 75.50 | 72.71 |
| | DoRA($r = 8$) | 5.16M | 85.06 | 93.00 | 89.57 | 57.16 | 27.17 | 76.40 | 71.39 |
| | MELoRA($r = 8$) | 4.82M | 86.84 | 90.56 | 90.35 | 58.86 | 30.18 | 74.53 | 71.89 |
| | HydraLoRA($r = 4$) | 5.94M | 85.82 | 91.06 | 91.27 | 58.68 | 29.13 | 74.47 | 71.74 |
| | **BoRA**($r = 8, b = 8$) | 4.86M | 87.93 | 93.22 | 90.35 | 58.91 | 29.66 | 75.37 | 72.57 |
| | **BoRA**($r = 8, b = 16$) | 4.99M | 87.85 | 92.50 | 90.94 | 59.72 | 31.36 | 76.20 | **73.10** |
| LLama-3-8B | LoRA($r = 8$) | 4.72M | 82.28 | 87.06 | 91.60 | 55.65 | 24.02 | 68.53 | 68.19 |
| | LoRA($r = 16$) | 9.44M | 84.56 | 91.22 | 92.26 | 57.22 | 25.72 | 70.17 | 70.19 |
| | LoRA($r = 32$) | 18.9M | 87.17 | 93.39 | 93.50 | 57.87 | 26.25 | 71.83 | 71.67 |
| | DoRA($r = 8$) | 4.92M | 81.39 | 89.09 | 92.42 | 55.77 | 23.63 | 68.15 | 68.41 |
| | MELoRA($r = 8$) | 4.72M | 85.32 | 85.67 | 91.34 | 54.74 | 20.87 | 70.90 | 68.14 |
| | HydraLoRA($r = 4$) | 5.11M | 77.22 | 89.17 | 91.73 | 56.63 | 24.41 | 66.30 | 67.58 |
| | **BoRA**($r = 8, b = 8$) | 4.77M | 87.85 | 93.67 | 92.72 | 58.45 | 26.38 | 70.10 | 71.53 |
| | **BoRA**($r = 8, b = 16$) | 4.92M | 88.35 | 93.00 | 92.72 | 58.83 | 27.17 | 73.40 | **72.24** |
| Qwen2.5-14B | LoRA($r = 8$) | 8.65M | 93.16 | 96.67 | 92.32 | 75.66 | 31.10 | 85.60 | 79.09 |
| | LoRA($r = 16$) | 17.3M | 91.90 | 96.33 | 92.91 | 74.37 | 34.65 | 86.40 | 79.43 |
| | LoRA($r = 32$) | 34.6M | 92.24 | 97.39 | 92.98 | 76.37 | 34.78 | 87.13 | 80.15 |
| | DoRA($r = 8$) | 8.99M | 92.91 | 96.72 | 91.86 | 75.26 | 33.73 | 86.13 | 79.44 |
| | MELoRA($r = 8$) | 8.65M | 91.65 | 97.17 | 92.32 | 75.66 | 33.46 | 84.90 | 79.19 |
| | HydraLoRA($r = 4$) | 9.29M | 92.74 | 96.78 | 91.60 | 75.76 | 33.33 | 86.23 | 79.41 |
| | **BoRA**($r = 8, b = 8$) | 8.72M | 91.65 | 96.83 | 92.78 | 75.41 | 34.78 | 86.93 | 79.73 |
| | **BoRA**($r = 8, b = 16$) | 8.95M | 91.90 | 98.00 | 92.72 | 75.82 | 38.19 | 87.00 | **80.60** |

increase in LoRA's rank is only 0.77%. This suggests that BoRA can achieve similar fine-tuning performance while requiring more than four times fewer parameters.

Table 3: The accuracy on **Commonsense Reasoning** tasks with various pretrained models.

| | | #Param | BoolQ | PIQA | SIQA | HellaSwag | WinoGrande | ARC-c | ARC-e | OBQA | Avg |
|---|---|---|---|---|---|---|---|---|---|---|---|
| Gemma-7B | LoRA($r = 8$) | 4.82M | 70.15 | 88.96 | 78.05 | 94.08 | 89.82 | 83.96 | 94.02 | 88.60 | 85.95 |
| | LoRA($r = 16$) | 9.63M | 75.17 | 88.74 | 77.58 | 95.21 | 89.11 | 84.73 | 92.93 | 88.00 | 86.43 |
| | LoRA($r = 32$) | 19.3M | 74.34 | 89.72 | 78.10 | 95.77 | 88.95 | 84.73 | 94.11 | 87.20 | 86.61 |
| | DoRA($r = 8$) | 5.16M | 74.86 | 89.55 | 78.76 | 93.66 | 89.74 | 83.45 | 92.68 | 86.20 | 86.11 |
| | MELoRA($r = 8$) | 4.82M | 71.22 | 88.90 | 78.97 | 94.46 | 88.32 | 83.36 | 93.64 | 87.20 | 85.76 |
| | HydraLoRA($r = 4$) | 5.94M | 71.77 | 87.92 | 80.76 | 95.00 | 88.24 | 84.47 | 94.44 | 87.20 | 86.23 |
| | **BoRA**($r = 8, b = 8$) | 4.86M | 72.66 | 90.26 | 78.97 | 94.77 | 90.77 | 84.73 | 93.39 | 89.00 | 86.82 |
| | **BoRA**($r = 8, b = 16$) | 4.99M | 73.08 | 90.42 | 80.04 | 95.10 | 89.82 | 84.95 | 94.36 | 88.53 | **87.04** |
| LLama-3-8B | LoRA($r = 8$) | 4.72M | 73.17 | 89.34 | 80.64 | 93.22 | 87.42 | 80.20 | 92.51 | 87.13 | 85.45 |
| | LoRA($r = 16$) | 9.44M | 73.54 | 89.50 | 81.18 | 94.18 | 88.00 | 81.11 | 93.15 | 88.53 | 86.15 |
| | LoRA($r = 32$) | 18.9M | 73.87 | 90.01 | 82.09 | 94.95 | 88.37 | 82.20 | 93.57 | 88.73 | **86.72** |
| | DoRA($r = 8$) | 4.92M | 72.72 | 89.17 | 80.76 | 93.51 | 88.32 | 80.89 | 92.34 | 88.00 | 85.71 |
| | MELoRA($r = 8$) | 4.72M | 72.66 | 89.28 | 81.53 | 94.19 | 86.74 | 80.89 | 93.10 | 88.20 | 85.92 |
| | HydraLoRA($r = 4$) | 5.11M | 72.66 | 88.85 | 81.01 | 93.48 | 87.37 | 80.20 | 92.68 | 87.80 | 85.51 |
| | **BoRA**($r = 8, b = 8$) | 4.77M | 74.10 | 89.61 | 81.68 | 93.81 | 88.24 | 80.97 | 93.01 | 88.40 | 86.23 |
| | **BoRA**($r = 8, b = 16$) | 4.92M | 74.22 | 89.66 | 81.93 | 94.27 | 88.48 | 82.08 | 93.10 | 89.20 | 86.62 |
| Qwen2.5-14B | LoRA($r = 8$) | 8.65M | 75.99 | 93.80 | 84.34 | 96.39 | 92.34 | 94.03 | 98.06 | 95.00 | 91.24 |
| | LoRA($r = 16$) | 17.3M | 76.15 | 93.74 | 84.44 | 96.87 | 92.50 | 94.20 | 98.36 | 95.80 | 91.51 |
| | LoRA($r = 32$) | 34.6M | 76.70 | 93.91 | 84.90 | 96.91 | 92.42 | 94.62 | 98.23 | 95.80 | 91.69 |
| | DoRA($r = 8$) | 8.99M | 76.35 | 93.49 | 84.32 | 96.56 | 92.66 | 94.34 | 98.13 | 95.73 | 91.45 |
| | MELoRA($r = 8$) | 8.65M | 76.41 | 93.67 | 84.87 | 96.78 | 91.92 | 94.25 | 98.01 | 95.93 | 91.48 |
| | HydraLoRA($r = 4$) | 9.29M | 76.02 | 93.91 | 84.24 | 96.50 | 92.58 | 94.11 | 97.98 | 95.20 | 91.32 |
| | **BoRA**($r = 8, b = 8$) | 8.72M | 77.09 | 93.85 | 84.75 | 96.95 | 92.27 | 94.62 | 98.32 | 96.00 | 91.73 |
| | **BoRA**($r = 8, b = 16$) | 8.95M | 77.06 | 93.91 | 85.57 | 96.96 | 92.66 | 93.52 | 98.11 | 96.80 | **91.82** |

## 4.3 ABLATION STUDIES

To optimize the block-wise diagonal matrices $\Sigma$ effectively, BoRA applies both an exponential and a normalization function to the learnable parameters $\sigma$, as shown in Eq.(4). In this section, we conduct ablation experiments on mathematical reasoning tasks to assess the importance of these two functions. As shown in Figure 3(a), omitting either the exponential or the normalization function leads to a significant decrease in accuracy across various models. Notably, the absence of normalization has a

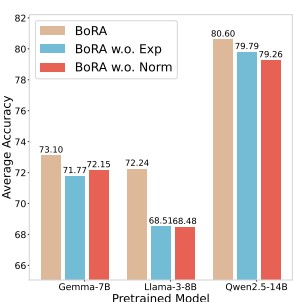 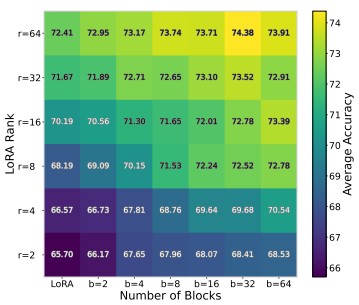 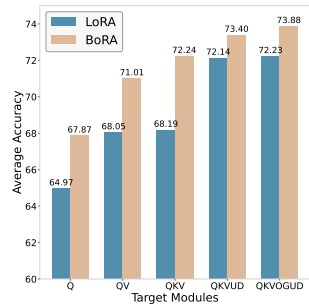

(a) Ablation Results of BoRA  (b) Various Ranks and Block Numbers  (c) Differnt Tuning Granularity

Figure 3: Ablation studies and scalability analysis on mathematical reasoning tasks. (a) Ablation results for the exponential function (Exp) and normalization function (Norm) in BoRA. (b) Accuracy of BoRA at varying ranks and block numbers. (c) Accuracy of BoRA at different tuning granularity. The figures present only the average accuracy, and the detailed results are available in Appendix C.

more significant effect on performance. This can be attributed to the small initial values of $\sigma$; without normalization, most entries in $\Sigma$ remain close to 1, which restricts the expressiveness of BoRA.

## 4.4 SCALABILITY ANALYSIS

In this section, we evaluate the scalability of BoRA on mathematical reasoning tasks using LLama-3-8B, from three perspectives: the LoRA rank, the number of blocks, and the tuning granularity.

**Results of Different LoRA Ranks and Number of Blocks.** In Section 4.2, we set the rank of BoRA to 8 and evaluated its performance with 8 and 16 blocks. In this section, we investigate the effects of varying the LoRA rank ($r$) and the number of blocks ($b$) from the set $\{2, 4, 8, 16, 32, 64\}$. Notably, the number of additional parameters introduced by BoRA is $b^2 r$, which increases quadratically with $b$. For comparison, when $b = 64$, the number of trainable parameters in BoRA is approximately 1.6 times greater than that in LoRA at the same rank. As shown in Figure 3(b), BoRA consistently outperforms LoRA across various ranks, even when $b = 2$. As $b$ increases, the performance of BoRA improves. However, when $r$ increases and $b$ surpasses a certain threshold, accuracy begins to decline, which is attributed to potential overfitting caused by the higher rank of the weight matrix.

**Results of Different Tuning Granularity.** Finally, we evaluate the scalability of BoRA under different tuning granularity. In Section 4.2, we apply LoRA and BoRA only to the query, key, and value weights. In this section, we introduce four additional strengths: Q, QV, QKVUD, and QKVOGUD, where O, G, U, and D represent the output, gate, up, and down projection weights, respectively. The rank is set to 8, and the number of blocks is set to 16 for BoRA. Note that, within any given tuning granularity, the number of parameters and computational costs in BoRA are approximately the same as in LoRA. The goal of this experiment is to compare BoRA's performance relative to LoRA at each granularity. As shown in Figure 3(c), BoRA consistently outperforms LoRA across different tuning granularity, demonstrating that BoRA's performance improvement over LoRA is consistent across various layers of the model (e.g., k_proj and v_proj).

## 4.5 SINGULAR VALUE ANALYSIS

BoRA aims to enhance the rank of LoRA weights. To illustrate this more clearly, we analyze the singular values of both LoRA and BoRA. Specifically, we present the sum of the squared singular values and the count of singular values exceeding 0.005 for the weight of each query layer. The threshold of 0.005 is chosen to calculate the effective rank, following the settings used in HiRA Huang et al. (2025). As shown in Figure 4, the number of singular values greater than 0.005 in LoRA corresponds to its rank. As the rank increases, the sum of squared singular values also increases. In contrast to LoRA, BoRA exhibits significantly more singular values greater than 0.005, and the sum

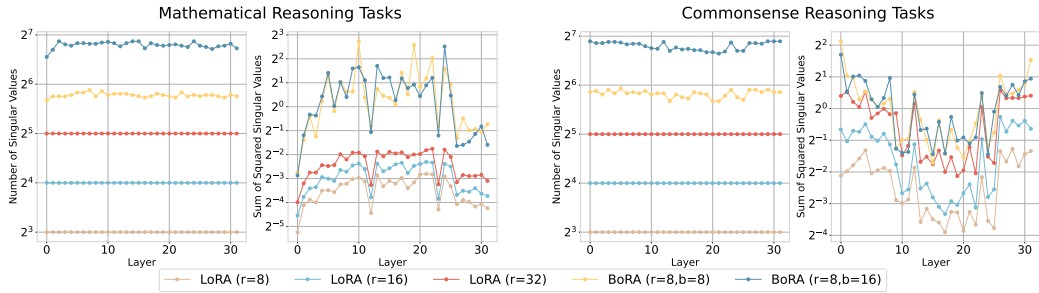

Figure 4: The sum of the squared singular values and the number of singular values greater than 0.005 across the query layers in LoRA and BoRA, with different ranks and block numbers.

of squared singular values is substantially larger, effectively demonstrating BoRA's role in enhancing rank.

## 4.6 EFFICIENCY ANALYSIS

As discussed in Section 3.4, we highlighted the minimal computational and memory overhead introduced by the block-wise diagonal matrices in BoRA. To validate this point more rigorously, we measured the wall-clock time and memory usage of various methods. One epoch of training on the mathematical reasoning task was performed using the LLama-3-8B model with an Nvidia A6000 GPU and a batch size of 4.

Table 4: Computational cost and memory usage of different methods using the LLama-3-8B model.

|  | # Param (M) | Memory (GB) | Time (Min) |
|---|---|---|---|
| LoRA($r = 8$) | 4.72 | 23.57 | 18.5 |
| BoRA($r = 8, b = 16$) | 4.92 | 23.63 | 18.6 |
| MELoRA($r = 8$) | 4.72 | 23.57 | 18.6 |
| HydraLoRA($r = 4$) | 5.11 | 24.98 | 19.6 |
| DoRA($r = 8$) | 4.92 | 24.36 | 22.3 |

As shown in Table 4, BoRA and MELoRA exhibit nearly identical training time and memory footprint compared to LoRA. HydraLoRA introduces additional gating layers, which add minor computational cost and result in a small increase in runtime. DoRA, due to its reparameterization of directions and magnitudes, requires more complex computation and therefore introduces some overhead as well. Overall, the empirical evidence supports our argument that the end-to-end training overhead of BoRA is essentially on par with LoRA, and comparable or lower than several other state-of-the-art methods.

## 5 CONCLUSION

In this paper, we analyze the rank of LoRA weights from the perspective of block matrix multiplication. Our analysis revealed that standard LoRA suffers from rank limitations due to correlations among different block products. To solve this limitation, we propose Block-Diversified Low-Rank Adaptation (BoRA), a simple yet powerful extension to LoRA that enhances the rank of LoRA weights. BoRA effectively breaks the correlations among different block products by introducing a unique diagonal matrix for each block multiplication, thereby increasing the rank of LoRA weights by a factor of $b$, where $b$ denotes the number of blocks. Extensive experiments demonstrate that BoRA outperforms LoRA and several LoRA variants across various tasks and models. Notably, BoRA achieves a 2-4% accuracy improvement over LoRA, while maintaining similar parameter and computational costs.

ACKNOWLEDGEMENTS

This work is supported by the National Natural Science Foundation of China under grants 62376103, 62302184, 62436003 and 62206102; Major Science and Technology Project of Hubei Province under grant 2025BAB011 and 2024BAA008; Hubei Science and Technology Talent Service Project under grant 2024DJC078; and Ant Group through CCF-Ant Research Fund. The computation is completed in the HPC Platform of Huazhong University of Science and Technology.

ETHICS STATEMENT

The authors confirm that the submitted work does not raise any concerns with respect to the ICLR Code of Ethics. It does not involve human subjects, sensitive or private data, or applications with potential ethical risks. All resources used are publicly available and properly licensed, and the research was conducted in full compliance with ethical and legal standards.

REPRODUCIBILITY STATEMENT

This paper includes detailed descriptions of the training setups, hyperparameter choices, and evaluation protocols, enabling full verification of the methodology. To further support reproducibility, we make our code publicly available at `https://github.com/Leopold1423/bora-iclr26`.

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

## A   LLM USAGE

Large language models (LLMs) were only used as writing assistants to polish the language, improve clarity, and check grammar. They were not involved in the generation of research ideas, the design or implementation of methods, data analysis, or the production of results. The authors take full responsibility for all content of the paper.

## B   DETAILED EXPERIMENTAL SETTINGS

**Datasets and Models.** The GLUE benchmark Wang et al. (2019) includes two single-sentence classification tasks (CoLA, SST-2), five pairwise text classification tasks (MNLI, RTE, QQP, MRPC, and QNLI), and one text similarity prediction task (STS-B). This paper reports the overall matched and mismatched accuracy for MNLI, Matthew's correlation for CoLA, Pearson correlation for STS-B, and accuracy for the remaining tasks. Due to the differing dataset sizes, the number of epochs varies: 10 epochs for RTE and MRPC, 5 epochs for STS-B and CoLA, 2 epochs for SST-2 and QNLI, and 1 epoch for MNLI and QQP. The models used are RoBERTa-Base and RoBERTa-Large Liu et al. (2019).

**LoRA Hyperparameters.** The scaling factor is set to $\alpha = 2r$, where $r$ is the LoRA rank. LoRA is applied to the query and value weights with a dropout rate of 0.05, using full precision (FP32).

**Training Hyperparameters.** AdamW Loshchilov and Hutter (2019) is used with $\beta_1 = 0.9$, $\beta_2 = 0.999$, $\epsilon = 1e{-}8$, and no weight decay. The learning rate is selected from the set $\{3e{-}5, 1e{-}4, 3e{-}4, 1e{-}3\}$, with optimal values of $3e{-}4$ for RoBERTa-Base and $1e{-}4$ for RoBERTa-Large. A warm-up ratio of 0.03 is applied, and the batch size is set to 32. The maximum sequence length is 512.

### B.1   EXPERIMENTS ON MATHEMATICAL AND COMMONSENSE REASONING TASKS

**Datasets.** Mathematical and commonsense reasoning tasks contain 10K and 170K training samples, respectively, along with several test tasks. Note that we directly utilize the data from Hu et al. (2023) for our experiments. The training process consists of a single epoch. Three models are employed: Gemma-7B Mesnard et al. (2024), LLama-3-8B Dubey et al. (2024), and Qwen2.5-14B Yang et al. (2024).

**LoRA Hyperparameters.** The scaling factor is set to $\alpha = 2r$, where $r$ is the LoRA rank. LoRA is applied to the query, key, and value weights with a dropout rate of 0.05, using half precision (BF16).

**Training Hyperparameters.** AdamW is employed with the same settings as previously mentioned. The learning rate is chosen from the set $\{3e{-}5, 1e{-}4, 3e{-}4, 1e{-}3\}$ and is set to $1e{-}4$. A warm-up of 100 steps is applied, and the batch size is set to 16. The maximum sequence length is 256.

## C   ADDITIONAL EXPERIMENTAL RESULTS

### C.1   STANDARD DEVIATIONS

As discussed in Section 4.2, each experiment was repeated three times, and the average results are reported. For conciseness, standard deviations are provided in the Appendix. Table 5 shows the standard deviations for each GLUE dataset, where a separate model was trained for each. Table 6 presents the standard deviation of the average accuracy for the commonsense and mathematical reasoning tasks, where a single model was used across these sub-tasks. Notably, the standard deviation remains stable and is much smaller than the accuracy improvement achieved by BoRA.

### C.2   DETAILED ABLATION RESULTS

In Section 4.3, ablation experiments were conducted on mathematical reasoning tasks to evaluate the contributions of exponential and normalization functions. Figure 3(a) shows the average accuracy of various methods across multiple sub-tasks. Full experimental results are provided in Table 7 for a more detailed comparison.

Table 5: The standard deviation of different methods on the GLUE benchmark.

| | | RTE | MRPC | STS-B | CoLA | SST-2 | QNLI | MNLI | QQP |
|---|---|---|---|---|---|---|---|---|---|
| | LoRA($r = 8$) | 0.51 | 0.46 | 0.27 | 0.57 | 0.11 | 0.15 | 0.11 | 0.10 |
| | LoRA($r = 16$) | 0.84 | 0.40 | 0.34 | 0.64 | 0.29 | 0.11 | 0.04 | 0.08 |
| | LoRA($r = 32$) | 0.78 | 0.35 | 0.39 | 0.56 | 0.14 | 1.00 | 0.11 | 0.03 |
| | DoRA($r = 8$) | 0.51 | 0.80 | 0.41 | 0.49 | 0.41 | 0.09 | 0.12 | 0.10 |
| RoBERTa-Base | MELoRA($r = 8$) | 0.85 | 0.72 | 0.76 | 0.68 | 0.09 | 0.30 | 0.30 | 0.09 |
| | HydraLoRA($r = 4$) | 0.18 | 0.53 | 0.51 | 0.55 | 0.24 | 0.15 | 0.27 | 0.05 |
| | BoRA($r = 8, b = 8$) | 0.86 | 0.12 | 0.61 | 0.52 | 0.22 | 0.02 | 0.27 | 0.04 |
| | BoRA($r = 8, b = 16$) | 0.64 | 0.61 | 0.88 | 0.64 | 0.19 | 0.20 | 0.14 | 0.12 |
| | LoRA($r = 8$) | 0.55 | 0.81 | 0.26 | 0.64 | 0.50 | 0.22 | 0.23 | 0.05 |
| | LoRA($r = 16$) | 0.51 | 0.20 | 0.59 | 0.98 | 0.19 | 0.16 | 0.27 | 0.04 |
| | LoRA($r = 32$) | 0.40 | 0.66 | 0.13 | 0.65 | 0.33 | 0.44 | 0.32 | 0.02 |
| | DoRA($r = 8$) | 0.79 | 0.42 | 0.03 | 0.78 | 0.19 | 0.30 | 0.25 | 0.02 |
| RoBERTa-Large | MELoRA($r = 8$) | 0.59 | 0.35 | 0.89 | 0.24 | 0.19 | 0.25 | 0.38 | 0.04 |
| | HydraLoRA($r = 4$) | 0.25 | 0.60 | 0.63 | 0.59 | 0.09 | 0.04 | 0.29 | 0.09 |
| | BoRA($r = 8, b = 8$) | 0.36 | 0.60 | 0.15 | 0.43 | 0.09 | 0.13 | 0.48 | 0.03 |
| | BoRA($r = 8, b = 16$) | 0.59 | 0.23 | 0.17 | 0.53 | 0.19 | 0.17 | 0.12 | 0.03 |

Table 6: The standard deviation of the average accuracy on the mathematical and commonsense reasoning tasks with various pretrained models.

| | Mathematical Reasoning | | | Commonsense Reasoning | | |
|---|---|---|---|---|---|---|
| | Gemma-7B | LLama-3-8B | Qwen2.5-14B | Gemma-7B | LLama-3-8B | Qwen2.5-14B |
| LoRA($r = 8$) | 0.19 | 0.88 | 0.60 | 0.63 | 0.13 | 0.13 |
| LoRA($r = 16$) | 0.61 | 0.62 | 0.50 | 0.22 | 0.19 | 0.10 |
| LoRA($r = 32$) | 0.48 | 0.80 | 0.78 | 0.18 | 0.29 | 0.06 |
| DoRA($r = 8$) | 0.47 | 0.47 | 0.49 | 0.48 | 0.35 | 0.17 |
| MELoRA($r = 8$) | 0.34 | 0.58 | 0.23 | 0.22 | 0.48 | 0.02 |
| HydraLoRA($r = 4$) | 0.21 | 0.36 | 0.39 | 0.43 | 0.19 | 0.23 |
| BoRA($r = 8, b = 8$) | 0.42 | 0.61 | 0.38 | 0.12 | 0.26 | 0.06 |
| BoRA($r = 8, b = 16$) | 0.44 | 0.38 | 0.32 | 0.38 | 0.28 | 0.01 |

## C.3 DETAILED RESULTS OF THE SCALABILITY ANALYSIS

In Section 4.4, the scalability of BoRA was evaluated using LLama-3-8B on mathematical reasoning tasks from three perspectives: LoRA rank, number of blocks, and tuning granularity. Figure 3 presents the average accuracy of different settings across sub-tasks. Complete experimental results are available in Tables 8 and 9 for further comparison.

## D BROADER IMPACTS

This paper introduces BoRA, a method for increasing the rank of LoRA weights. BoRA achieves the same fine-tuning performance as LoRA while using fewer parameters and reducing computational overhead, thus contributing to energy savings. Our work builds upon LoRA, and we assert that our approach does not introduce any negative social implications requiring further discussion.

## E ADDITIONAL RELATED WORK

In addition to LoRA, there are two other commonly used PEFT methods: adapter-based and soft prompt-based methods. The adapter-based method Houlsby et al. (2019); He et al. (2022); Wang et al. (2022) inserts new layers into the model and fine-tunes only these layers, significantly reducing resource consumption. However, the additional layers introduce increased latency. The soft prompt-based method Lester et al. (2021); Li and Liang (2021); Razdaibiedina et al. (2023) adds learnable soft tokens (prompts) to the input, enabling the model to adapt to specific tasks. This method leverages the pretrained model's inherent capabilities and requires only appropriate prompts for downstream task adaptation. However, it also adds computational overhead and increases inference latency. In

Table 7: The accuracy of BoRA without exponential or normalization functions on mathematical reasoning tasks using LLama-3-8B.

| | | AddSub | MultiArith | SingleEq | GSM8K | AQuA | SVAMP | Avg |
|---|---|---|---|---|---|---|---|---|
| | BoRA | 87.85 | 92.50 | 90.94 | 59.72 | 31.36 | 76.20 | 73.10 |
| Gemma-7B | BoRA w.o. Exp | 86.33 | 90.67 | 89.37 | 58.45 | 30.71 | 75.10 | 71.77 |
| | BoRA w.o. Norm | 86.58 | 91.83 | 88.78 | 58.23 | 32.28 | 75.20 | 72.15 |
| | BoRA | 88.35 | 93.00 | 92.72 | 58.83 | 27.17 | 73.40 | 72.24 |
| LLama-3-8B | BoRA w.o. Exp | 83.04 | 87.00 | 90.94 | 55.19 | 26.77 | 68.10 | 68.51 |
| | BoRA w.o. Norm | 81.77 | 88.17 | 91.34 | 56.10 | 25.20 | 68.30 | 68.48 |
| | BoRA | 91.90 | 98.00 | 92.72 | 75.82 | 38.19 | 87.00 | 80.60 |
| Qwen2.5-14B | BoRA w.o. Exp | 92.66 | 96.67 | 92.52 | 77.63 | 33.86 | 85.40 | 79.79 |
| | BoRA w.o. Norm | 93.16 | 96.17 | 92.13 | 76.42 | 32.28 | 85.40 | 79.26 |

Table 8: The accuracy of LoRA and BoRA with varying target modules on mathematical reasoning tasks using LLama-3-8B.

| Target Modules | Method | #Params | AddSub | MultiArith | SingleEq | GSM8K | AQuA | SVAMP | Avg |
|---|---|---|---|---|---|---|---|---|---|
| Q | LoRA($r = 8$) | 2.10M | 74.43 | 86.39 | 88.45 | 52.26 | 26.64 | 61.67 | 64.97 |
| | BoRA($r = 8, b = 16$) | 2.16M | 81.18 | 90.78 | 90.42 | 54.54 | 23.23 | 67.10 | 67.87 |
| QV | LoRA($r = 8$) | 3.41M | 81.27 | 90.00 | 91.54 | 56.25 | 22.83 | 66.40 | 68.05 |
| | BoRA($r = 8, b = 16$) | 3.54M | 86.50 | 91.22 | 93.18 | 58.18 | 25.98 | 71.03 | 71.01 |
| QKV | LoRA($r = 8$) | 4.72M | 82.28 | 87.06 | 91.60 | 55.65 | 24.02 | 68.53 | 68.19 |
| | BoRA($r = 8, b = 16$) | 4.92M | 88.35 | 93.00 | 92.72 | 58.83 | 27.17 | 73.40 | 72.24 |
| QKVUD | LoRA($r = 8$) | 14.2M | 88.69 | 92.11 | 93.83 | 59.79 | 24.93 | 73.50 | 72.14 |
| | BoRA($r = 8, b = 16$) | 14.5M | 89.11 | 95.25 | 93.60 | 61.79 | 24.41 | 76.20 | 73.40 |
| QKVOGUD | LoRA($r = 8$) | 21.0M | 87.59 | 93.28 | 93.90 | 59.41 | 24.15 | 75.03 | 72.23 |
| | BoRA($r = 8, b = 16$) | 21.4M | 89.62 | 95.67 | 94.00 | 62.47 | 25.40 | 76.10 | 73.88 |

contrast, both LoRA and the proposed BoRA allow for manual integration of weight updates into pretrained weights after fine-tuning, avoiding additional inference latency.

## F    THEORETICAL JUSTIFICATION FOR PROPOSITION 1

In Section 3.3, we briefly discussed the theoretical justification for Proposition 1. Here, we provide a more detailed expansion and explain under which assumptions BoRA's rank can achieve the upper bound presented in Proposition 1.

Without loss of generality, for a pre-trained weight $W_0 \in \mathbb{R}^{m \times n}$, LoRA maintains two low-rank matrices $B \in \mathbb{R}^{m \times r}$ and $A \in \mathbb{R}^{r \times n}$. Let matrix $B \in \mathbb{R}^{m \times r} = [B_1, B_2, \ldots, B_b]^\top$ be divided into $b$ blocks, where $B_i \in \mathbb{R}^{\frac{m}{b} \times r}$. Similarly, matrix $A \in \mathbb{R}^{r \times n} = [A_1, A_2, \ldots, A_b]$ is divided into $b$ blocks, where $A_i \in \mathbb{R}^{r \times \frac{n}{b}}$. The LoRA weights generated by the proposed BoRA can be denoted as follows:

$$\Delta W = \begin{bmatrix} B_1 \Sigma_{1,1} A_1 & B_1 \Sigma_{1,2} A_2 & \ldots & B_1 \Sigma_{1,b} A_b \\ B_2 \Sigma_{2,1} A_1 & B_2 \Sigma_{2,2} A_2 & \ldots & B_2 \Sigma_{2,b} A_b \\ \vdots & \vdots & \vdots & \vdots \\ B_b \Sigma_{b,1} A_1 & B_b \Sigma_{b,2} A_2 & \ldots & B_b \Sigma_{b,b} A_b \end{bmatrix}$$

where $\Sigma_{i,j} \in \mathbb{R}^{r \times r}$ ($\forall i, j \in [b]$) are diagonal matrices. Note that $\Delta W$ can also be represented as the product of three matrices ($B' \in \mathbb{R}^{m \times br}$, $\Sigma' \in \mathbb{R}^{br \times br}$, and $A' \in \mathbb{R}^{br \times n}$), as shown below:

$$\Delta W = B' \Sigma' A' = \begin{bmatrix} B_1 & 0 & \ldots & 0 \\ 0 & B_2 & \ldots & 0 \\ \vdots & \vdots & \vdots & \vdots \\ 0 & 0 & \ldots & B_b \end{bmatrix} \begin{bmatrix} \Sigma_{1,1} & \Sigma_{1,2} & \ldots & \Sigma_{1,b} \\ \Sigma_{2,1} & \Sigma_{2,2} & \ldots & \Sigma_{2,b} \\ \vdots & \vdots & \vdots & \vdots \\ \Sigma_{b,1} & \Sigma_{b,2} & \ldots & \Sigma_{b,b} \end{bmatrix} \begin{bmatrix} A_1 & 0 & \ldots & 0 \\ 0 & A_2 & \ldots & 0 \\ \vdots & \vdots & \vdots & \vdots \\ 0 & 0 & \ldots & A_b \end{bmatrix}$$

Based on the shapes of the matrices, we can derive the upper bound for the rank of the three matrices:

$$\text{rank}(B') \leq \min\{m, br\}, \ \text{rank}(A') \leq \min\{n, br\}, \ \text{rank}(\Sigma') \leq br$$

Table 9: The accuracy of LoRA and BoRA with varying ranks and block numbers on mathematical reasoning tasks using LLama-3-8B.

| Rank | Method | #Params | AddSub | MultiArith | SingleEq | GSM8K | AQuA | SVAMP | Avg |
|------|--------|---------|--------|------------|----------|-------|------|-------|-----|
| | LoRA | 1.18M | 77.64 | 85.56 | 89.24 | 52.87 | 23.36 | 65.53 | 65.70 |
| | BoRA($b = 2$) | 1.18M | 74.94 | 89.17 | 88.19 | 56.03 | 26.77 | 61.90 | 66.17 |
| | BoRA($b = 4$) | 1.18M | 80.64 | 88.75 | 90.36 | 54.93 | 24.61 | 66.65 | 67.65 |
| $r = 2$ | BoRA($b = 8$) | 1.19M | 79.37 | 88.67 | 90.06 | 56.94 | 25.59 | 67.10 | 67.96 |
| | BoRA($b = 16$) | 1.23M | 81.86 | 87.33 | 91.73 | 55.34 | 24.67 | 67.47 | 68.07 |
| | BoRA($b = 32$) | 1.38M | 80.76 | 88.67 | 91.14 | 55.42 | 26.77 | 67.70 | 68.41 |
| | BoRA($b = 64$) | 1.97M | 84.30 | 87.83 | 90.75 | 56.33 | 22.83 | 69.10 | 68.53 |
| | LoRA | 2.36M | 79.49 | 86.83 | 88.85 | 55.12 | 23.10 | 66.03 | 66.57 |
| | BoRA($b = 2$) | 2.36M | 79.24 | 88.67 | 90.16 | 55.88 | 22.05 | 64.40 | 66.73 |
| | BoRA($b = 4$) | 2.37M | 79.24 | 89.50 | 93.90 | 58.30 | 22.05 | 63.90 | 67.81 |
| $r = 4$ | BoRA($b = 8$) | 2.38M | 83.29 | 90.17 | 91.53 | 55.55 | 24.28 | 67.73 | 68.76 |
| | BoRA($b = 16$) | 2.46M | 82.95 | 90.56 | 92.92 | 57.70 | 24.54 | 69.17 | 69.64 |
| | BoRA($b = 32$) | 2.75M | 81.60 | 91.39 | 92.78 | 58.20 | 25.98 | 68.10 | 69.68 |
| | BoRA($b = 64$) | 3.93M | 85.06 | 92.33 | 92.91 | 56.68 | 27.04 | 69.23 | 70.54 |
| | LoRA | 4.72M | 82.28 | 87.06 | 91.60 | 55.65 | 24.02 | 68.53 | 68.19 |
| | BoRA($b = 2$) | 4.72M | 81.01 | 92.33 | 91.34 | 59.36 | 24.41 | 66.10 | 69.09 |
| | BoRA($b = 4$) | 4.73M | 84.30 | 91.39 | 92.78 | 58.10 | 24.67 | 69.67 | 70.15 |
| $r = 8$ | BoRA($b = 8$) | 4.77M | 87.85 | 93.67 | 92.72 | 58.45 | 26.38 | 70.10 | 71.53 |
| | BoRA($b = 16$) | 4.92M | 88.35 | 93.00 | 92.72 | 58.83 | 27.17 | 73.40 | 72.24 |
| | BoRA($b = 32$) | 5.51M | 87.34 | 95.17 | 92.91 | 58.53 | 27.56 | 73.60 | 72.52 |
| | BoRA($b = 64$) | 7.86M | 87.34 | 94.33 | 94.09 | 60.35 | 27.95 | 72.60 | 72.78 |
| | LoRA | 9.44M | 84.56 | 91.22 | 92.26 | 57.22 | 25.72 | 70.17 | 70.19 |
| | BoRA($b = 2$) | 9.44M | 83.04 | 91.67 | 94.69 | 59.59 | 26.38 | 68.00 | 70.56 |
| | BoRA($b = 4$) | 9.46M | 86.16 | 91.33 | 93.83 | 59.26 | 26.51 | 70.70 | 71.30 |
| $r = 16$ | BoRA($b = 8$) | 9.54M | 85.82 | 93.61 | 93.31 | 59.61 | 25.46 | 72.10 | 71.65 |
| | BoRA($b = 16$) | 9.83M | 86.75 | 93.67 | 93.90 | 58.94 | 26.51 | 72.27 | 72.01 |
| | BoRA($b = 32$) | 11.0M | 89.37 | 94.83 | 94.29 | 58.91 | 25.20 | 74.10 | 72.78 |
| | BoRA($b = 64$) | 15.7M | 87.34 | 95.58 | 94.29 | 59.40 | 30.52 | 73.20 | 73.39 |
| | LoRA | 18.9M | 87.17 | 93.39 | 93.50 | 57.87 | 26.25 | 71.83 | 71.67 |
| | BoRA($b = 2$) | 18.9M | 87.85 | 91.83 | 94.29 | 59.06 | 25.98 | 72.30 | 71.89 |
| | BoRA($b = 4$) | 18.9M | 90.13 | 93.56 | 93.24 | 59.59 | 25.20 | 74.57 | 72.71 |
| $r = 32$ | BoRA($b = 8$) | 19.1M | 89.11 | 93.67 | 93.70 | 60.42 | 24.02 | 75.00 | 72.65 |
| | BoRA($b = 16$) | 19.7M | 88.61 | 95.83 | 93.50 | 59.59 | 27.56 | 73.50 | 73.10 |
| | BoRA($b = 32$) | 22.0M | 89.50 | 95.42 | 93.60 | 59.93 | 27.36 | 75.30 | 73.52 |
| | BoRA($b = 64$) | 31.5M | 89.62 | 96.67 | 92.91 | 60.65 | 23.62 | 74.00 | 72.91 |
| | LoRA | 37.7M | 88.10 | 94.33 | 93.70 | 60.05 | 25.20 | 73.10 | 72.41 |
| | BoRA($b = 2$) | 37.8M | 89.62 | 91.17 | 94.29 | 60.73 | 25.59 | 76.30 | 72.95 |
| | BoRA($b = 4$) | 37.8M | 88.77 | 94.72 | 93.70 | 60.78 | 26.51 | 74.53 | 73.17 |
| $r = 64$ | BoRA($b = 8$) | 38.1M | 90.13 | 94.50 | 94.49 | 62.70 | 25.20 | 75.40 | 73.74 |
| | BoRA($b = 16$) | 39.3M | 89.87 | 94.83 | 94.29 | 60.50 | 26.77 | 76.00 | 73.71 |
| | BoRA($b = 32$) | 44.0M | 89.62 | 96.17 | 95.08 | 61.56 | 27.95 | 75.90 | 74.38 |
| | BoRA($b = 64$) | 62.9M | 90.13 | 95.67 | 93.90 | 61.26 | 25.59 | 76.90 | 73.91 |

According to the properties of matrix multiplication:

$$\text{rank}(\Delta W) \leq \min\{\text{rank}(A'), \text{rank}(\Sigma'), \text{rank}(B')\} = \min\{m, n, br\}.$$

It is important to note that $\text{rank}(\Delta W)$ can only have a clearly defined upper bound, with the minimum value being 0. For example, in BoRA, the matrix $B$ is initialized to zero, so at the beginning of training, $\text{rank}(B') = 0$, which results in $\text{rank}(\Delta W) = 0$. However, when all three matrices are full rank (i.e., $\text{rank}(B') = \min\{m, br\}$, $\text{rank}(A') = \min\{n, br\}$, and $\text{rank}(\Sigma') = br$), equality holds, and the upper bound for $\text{rank}(\Delta W)$ can be achieved. This condition is typically satisfied because $\Sigma'$ and $A'$ are initialized randomly, which usually ensures that $\text{rank}(A') = \min\{n, br\}$ and $\text{rank}(\Sigma') = br$. As training progresses, $B'$ is continually updated, and $\text{rank}(B')$ can gradually approach full rank. This is evident from Figure 4 in the paper, where the effective rank of BoRA (the number of singular values greater than 0.005) approaches $br$.

