# OpenReview forum: "BoRA: Towards More Expressive Low-Rank Adaptation with Block Diversity"
_ICLR.cc/2026/Conference — ICLR 2026 Poster_

### Official Review · Reviewer_rMns · 2025-10-16

**Soundness:** 3
**Presentation:** 3
**Contribution:** 2
**Rating:** 4
**Confidence:** 3

**Summary:**

This paper proposes BoRA, a parameter-efficient fine-tuning method designed to enhance the expressiveness of standard LoRA. The core idea is to partition the LoRA matrices A and B into multiple blocks and introduce a unique learnable diagonal matrix $\Sigma$ for each block pair. Through extensive experiments, the authors demonstrate that BoRA consistently outperforms standard LoRA and its variants under a similar number of parameters.

**Strengths:**

1. The idea to improve LoRA's power of expressiveness by incorporating blockwise learnable diagonal matrices is novel and makes sense to me.
2. The experimental results consistently show the advantage of the proposed method.
3. The presentation is clear, and the paper is easy to follow.

**Weaknesses:**

1. I think the comparison with other LoRA-like baseline algorithms is not enough. Beyond LoRA, DoRA, MELoRA and HydraLoRA, there are other similar methods that also aim to improve LoRA's expressive power yet not compared or discussed in this paper. For example, 1) ReLoRA is discussed but not compared with; 2) SLTrain [https://arxiv.org/abs/2406.02214] is neither compared nor discussed, it can also largely improve the model performance; 3) LoLDU [https://arxiv.org/abs/2410.13618] is neither compared nor discussed, I believe its idea has some similarity with this paper.
2. The paper analyzes the FLOPs for the forward pass in Section 3.4, but does not discuss the actual overhead of backward propagation and end-to-end training time. Even if FLOPs are similar, introducing numerous small block operations may lead to increased kernel launch of CUDA, which could potentially result in a large amount of overhead. It is recommended to have a further discussion on the computation overhead.
3. A minor question: I notice that a prior work [https://arxiv.org/abs/2407.15857] in this literature has already used the name *BoRA* for their method. I recommend using another name to avoid ambiguity, or the authors can include some discussions about that BoRA algorithm.

**Questions:**

1. In practical fine-tuning, what is the percentage increase in time per training step for BoRA compared to standard LoRA or other baselines?
2. Can the authors provide more insights on how to decide an appropriate $b$ when using BoRA in practice, where both model scales and targeted tasks can vary?

---

> ### Author Response · Authors · 2025-11-21
> **Response to Reviewer rMns [Part I]**
>
> **Hi, Reviewer rMns:**
>
> Thank you for acknowledging the **novelty** of our idea and the **clarity** of our presentation, as well as the **consistent empirical improvements** demonstrated across benchmarks.
>
> We note that your primary concerns focus on the **comparison with additional baselines** and the **end-to-end training time**. Below, we address each of these points in detail.
>
> ***`Q1: "the comparison with other LoRA-like baseline algorithms"`***
>
> **R1:**
> Based on your suggestions, we have added comparisons with several additional baselines, including **HiRA** (Huang et al., 2025), **KronA** (Edalati et al., 2023), **LoLDU** (Shi et al., 2024) and **ReLoRA** (Lialinetal., 2024). Below, we report the average accuracy of various methods on the mathematical reasoning task using Llama-3-8B. The complete results will be included in the paper.
>
> | LLama-3-8B         | #Params | AddSub | MultiArith | SingleEq | gsm8k | AQuA  | SVAMP | Avg    |
> |---|---|---|---|---|---|---|---|---|
> | LoRA($r=8$)        | 4.72M | 82.28  | 87.06 | 91.60 | 55.65 | 24.02 | 68.53 | 68.19  |
> | HiRA($r=8$)        | 4.72M | 82.93  | 89.65 | 92.40 | 57.47 | 26.51 | 69.87 | 69.81  |
> | BoRA($r=8,b=16$)   | 4.92M | 88.35  | 93.00 | 92.72 | 58.83 | 27.17 | 73.40 | 72.24  |
> | KronA              | 0.52M | 77.97  | 84.33 | 87.40 | 48.82 | 25.59 | 62.70 | 64.47  |
> | BoRA+($r=8,b=16$)  | 0.20M | 84.56  | 88.67 | 90.94 | 53.48 | 25.77 | 67.10 | 68.42  |
> | BoRA+($r=16,b=16$) | 0.39M | 85.06  | 90.17 | 91.34 | 53.31 | 25.44 | 67.60 | 68.82  |
> | LoLDU              | 0.05M | 71.39  | 78.89 | 82.29 | 40.47 | 18.25 | 53.79 | 57.51  |
> | BoRA+($r=2,b=16$)  | 0.05M | 76.71  | 82.83 | 86.81 | 45.56 | 23.62 | 59.8  | 62.56  |
> | ReLoRA($r=8$)      | 4.72M | 83.29  | 89.5 | 93.31 | 56.1  | 25.2  | 72.5  | 69.98  |
> | ReBoRA($r=8,b=16$) | 4.92M | 88.85  | 95.5 | 92.93 | 58.48 | 28.74 | 73.9  | 73.07  |
>
> * **HiRA** increases the rank of LoRA weights by employing the Hadamard product with the formulation $\Delta W = W_0 \odot (BA)$, where $W_0$ is the pre-trained weight. As shown above, HiRA ($r=8$) indeed achieves slightly better accuracy than LoRA ($r=8$); however, its performance still falls noticeably short compared to BoRA ($r=8,b=16$) under a similar parameter budget.
>
> * Similarly, **KronA** increases the rank of LoRA weights by employing the Kronecker product with the formulation $\Delta W = B \otimes A \in \mathbb{R}^{m \times n}$. Here, $B \in \mathbb{R}^{b_1 \times b_2}$, $A \in \mathbb{R}^{a_1 \times a_2}$, and $a_1 b_1 = m$, $a_2 b_2 = n$. Following the original paper, we set $a_1$ and $b_1$, as well as $a_2$ and $b_2$, to be as close as possible to minimize the number of parameters. As a result, KronA has significantly fewer parameters than LoRA and BoRA. To compare fairly with KronA and inspired by **Reviewer dURX**, we constructed a variant of BoRA by freezing matrices $A$ and $B$, applying PiSSA (Meng et al., 2024) to initialize them (via SVD on the pre-trained weight), and initializing the learnable block-wise diagonal parameters $\sigma$ to zero. We denote this variant as **BoRA+**. As shown above, BoRA+ ($r=8,b=16$) matches KronA in parameter count but achieves substantially higher accuracy.
>
> * **LoLDU** decomposes pre-trained weights using LDU (Lower-Diagonal-Upper) decomposition as $W_0 = LDU$, where $D$ is a diagonal matrix. It enables efficient adaptation by training only the top-$r$ diagonal matrix $D_r$ while keeping the other components (L, U) fixed. While both LoLDU and BoRA involve diagonal matrices, our key innovation lies in the use of block-wise diagonal matrices, which break inter-block dependencies and increase the rank. It is important to note that LoLDU typically has very few trainable parameters, so we compare it to the BoRA+ variant discussed earlier. As shown above, with the similar parameter count, BoRA+ ($r=2,b=16$) consistently outperforms LoLDU ($r=512$). Furthermore, although LoLDU has fewer trainable parameters, the dimensions of its $A$ and $B$ matrices are relatively large, leading to higher computational costs, a point we explore further in the next question.
>
> * **SLTrain** targets low-rank training during pretraining, which differs from our fine-tuning setting and is therefore not directly comparable in our experimental setup. Nonetheless, your comment indeed reminds us to discuss the role of low-rank methods in pretraining more explicitly in the related work section.
>
> * Finally, **ReLoRA** improves performance by periodically merging the low-rank weights back into pre-trained weights. Since ReLoRA does not alter the LoRA structure itself, it is orthogonal to our BoRA method. We therefore combine the two and refer to the hybrid approach as **ReBoRA**. Results show that our method can also enhance ReLoRA’s performance while using only a very small number of additional parameters.
>
> `Due to space limitations, please refer to Part II for our responses to other questions.`

---

> ### Author Response · Authors · 2025-11-21
> **Response to Reviewer rMns [Part II]**
>
> ***`Q2: "the actual overhead of end-to-end training"`***
>
> **R2:** Typically, GEMMs (General Matrix Multiplications) on GPUs utilize **Tiling** techniques, where large matrix multiplications are divided into smaller blocks to leverage parallelism and optimize memory access within the GPU's architecture.
> Our BoRA method partitions both the input and the low-rank matrices in a manner consistent with this technique. As a result, the partitioning process does not introduce any noticeable latency.
>
> To more rigorously support this claim, we measured the wall-clock time and memory usage of various methods. One epoch of training on the mathematical reasoning task was performed using the LLAMA-3-8B model, with an Nvidia A6000 GPU and a batch size of 4.
>
> |                  | # Param (M) | Memory (GB) | Time (Min)  |
> |------------------|-------------|-------------|-------------|
> | LoRA($r=8$)      | 4.72        | 23.57       | 18.5        |
> | BoRA($r=8,b=16$) | 4.92        | 23.63       | 18.6        |
> | MELoRA($r=8$)    | 4.72        | 23.57       | 18.6        |
> | HiRA($r=8$)      | 4.72        | 23.57       | 18.5        |
> | KronA($r=8$)     | 0.52        | 25.03       | 18.9        |
> | ReLoRA($r=8$)    | 4.72        | 23.57       | 18.7        |
> | HydraLoRA($r=4$) | 5.11        | 24.98       | 19.6        |
> | LoLDU($r=512$)   | 0.05        | 23.69       | 20.1        |
> | DoRA($r=8$)      | 4.92        | 24.36       | 22.3        |
>
>
> The results show the following:
>
> * **BoRA**, **MELoRA** and **HiRA** exhibit nearly identical training time and memory footprint compared to LoRA.
>
> * **KronA**, despite reducing a substantial number of parameters using Kronecker products, does not show noticeable training-time speedup. This behavior is also confirmed in the original paper (see Table 2 in [https://arxiv.org/pdf/2212.10650]).
>
> * **ReLoRA** performs a few temporary resets during training, but the associated overhead is negligible.
>
> * **HydraLoRA** introduces additional gating layers, which add minor computational cost and result in a small increase in runtime.
> * **LoLDU**, although having fewer trainable parameters, uses relatively large A and B matrices, increasing computation and leading to slightly longer training time.
> * **DoRA**, due to its reparameterization of directions and magnitudes, requires more complex computation and therefore introduces some overhead as well.
>
> Overall, the empirical evidence supports our argument that the end-to-end training overhead of BoRA is essentially **on par** with LoRA, and **comparable or lower** than several other state-of-the-art methods. These results will be included in the paper.
>
>
> ***`Q3: "the naming conflict with a prior method also called BoRA"`***
>
> **R3:**
> Thank you for the suggestion.
> The prior method named BoRA [https://arxiv.org/abs/2407.15857] refers to a Bayesian hierarchical **multi-task fine-tuning** approach that shares information across tasks through global hierarchical priors.
> Our method, Block-Diversified Low-Rank Adaptation, by contrast, is a **structural enhancement** of LoRA and is conceptually different from that work.
>
> To avoid any naming conflict, we will rename our method to **BoDA** (**B**l**o**ck-**D**iversified Low-Rank **A**daptation) in future versions.
> For the current review cycle, however, we retain the name BoRA to avoid potential confusion among other reviewers.
>
>
> ***`Q4 "how to decide an appropriate $b$ when using BoRA in practice"`***
>
> **R4:**
> Thank you for raising this insightful question.
> Indeed, there is a useful trick for choosing the value of $b$.
>
> To achieve a target rank $R=br$, the number of parameters required by BoRA is $𝑁=(𝑚+𝑛+𝑏^2)𝑟=𝑅(\frac{𝑚+𝑛}{b}+𝑏)$,
> and this quantity is minimized when $b=\sqrt{m+n}$.
> From the perspective of parameter efficiency, this suggests that setting $b\approx \sqrt{m+n}$ is a good practical choice.
>
> This optimal value arises from a natural trade-off: the achievable rank of BoRA (i.e., $br$) increases linearly with $b$, while the additional parameter cost (i.e., $b^2r$) grows quadratically. Once $b$ exceeds $\sqrt{m+n}$, further increasing $b$ to obtain a higher rank becomes less efficient than simply increasing the base rank $r$ directly.
>
> Therefore, in practice, **one can simply set $b=\left\lfloor \sqrt{n} \right\rfloor$**. For instance, Llama-3–8B has a hidden size of 4096 (i.e., $n=4096$), so choosing $b=32$ works well.
> These discussions will be included in the paper.
>
> **`Lastly, thank you again for your valuable feedback, which has greatly improved our work. We hope our responses address your concerns and clarify any doubts. Please feel free to reach out to us if you have any further questions.`**

---

> > ### Comment · Reviewer_rMns · 2025-11-23
> > **Thank you for the response.**
> >
> > Thank you for the detailed responses regarding my questions. I'm satisfied with the reply and will increase my rating accordingly.

---

> > > ### Author Response · Authors · 2025-11-23
> > > **Sincere Thanks for Your Support**
> > >
> > > Thank you very much for your positive update, and once again, we really appreciate your time and the helpful suggestions you provided for our work.

---

### Official Review · Reviewer_9WMS · 2025-10-19

**Soundness:** 2
**Presentation:** 3
**Contribution:** 3
**Rating:** 4
**Confidence:** 3

**Summary:**

The paper presents Block-Diversified Low-Rank Adaptation (BoRA) to enhance rank by introducing block-wise diagonal matrices.
This increases rank with only a few additional parameters. Theoretical analysis confirms provable rank enhancement with an upper bound.
Empirical results show that BoRA achieves 2-4% accuracy improvements over LoRA across GLUE, mathematical, and commonsense reasoning tasks with similar parameter counts and involves various network architectures.

**Strengths:**

1. The paper introduces a novel perspective on LoRA by analyzing it through block matrix multiplication, revealing how correlations between block products constrain rank.
2. The paper demonstrates that both standard LoRA and MELoRA are special cases of BoRA, creating a unified theoretical framework.
3. Empirical results are solid, and BoRA is compared with various base models and tasks while achieving strong performance over a range of baselines.
4. The paper provides thorough ablation studies that validate the importance of both the exponential function and normalization function in BoRA. Results also show the consistently superior performance with varying ranks and block numbers.

**Weaknesses:**

1. In Section 3.2, the authors states
> Assuming $A$ and $B$ are divided into $b$ blocks along columns and rows, respectively, BoRA will additionally learn a set of diagonal matrices $\\{\Sigma_{i, j} \in \mathbb{R}^{r \times r} \mid i, j \in[r]\\}$.

​	This notation shows $i, j \in [r]$ for the indices of $\Sigma_{i,j}$, but this is inconsistent with the block count $b$. Since $A$ and $B$ are divided into $b$ blocks, the indices should be $i, j \in [b]$, not $[r]$.

2. Proposition 1 claims that BoRA requires only $b^{2} r$ additional parameters to increase the weight rank by a factor of $b$. In contrast, LoRA needs $(m+n)(b^{2} r)$ parameters to achieve the same bound, which, however, is incorrect. To achieve a rank of $br$, standard LoRA would need to set its rank parameter to $br$, which would require $r'(m+n)$ parameters where $r'=br$, thus $br(m+n)$ parameters - not $(m+n)(b^{2}r)$. A correct comparison is thus  LoRA (rank $br$) with $(m+n)br$ parameters vs. BoRA (rank bound $br$) with $(m+n)r + b^{2}r$ parameters. The authors should compare these two terms in practice.

3. Proposition 1 claims that the rank of BoRA's weight update $\Delta W$ satisfies $\operatorname{rank}(\Delta W) \leq \min \\{m, n, b r\\}$, but this bound is not properly justified and is too loose. Proposition 1 only gives $\operatorname{rank}(\Delta W) \leq b r$. The key claim that BoRA increases the rank by a factor of $b$ (i.e., up to $b r$) requires a tighter argument that BoRA can actually achieve this bound, which is not provided.

**Questions:**

1. Could you provide a theoretical justification that the rank of BoRA  can achieve $br$ under some assumptions, rather than merely being bounded by it?
2. In Section 4.5, why is $0.005$ chosen as the threshold?

---

> ### Author Response · Authors · 2025-11-21
> **Response to Reviewer 9WMS [Part I]**
>
> **Hi, Reviewer 9WMS:**
>
> Thank you for acknowledging the **novelty** of our idea and the **unified theoretical framework** that BoRA has created, as well as the **solid empirical results** and **thorough ablation studies**.
>
> We note that your primary concerns arise from **two minor typos** in Lines 178 and 213, as well as questions regarding **whether BoRA can achieve the rank upper bound stated in Proposition 1**. Below, we address each of these points in detail.
>
> ***`Q1: "typo in Line 178, where $i,j \in [r]$ should be $i,j \in [b]$"`***
>
> **R1:** Thank you for your thorough review.
> We have corrected the typo in **Line 178** (from $i,j \in [r]$ to $i,j \in [b]$), and the updated PDF will be uploaded in the next few days.
>
>
> ***`Q2: "typo in Line 213, where $(m+n)b^2r$ should be $(m+n)br$`***
>
> **R2:**
> Thank you for your careful and thorough review.
> Indeed, the expression in **Line 213** contains a typo: the number of parameters for LoRA to achieve a rank upper bound of $br$ should be $(m+n)br$, but we mistakenly wrote it as $(m+n)b^{2}r$. We have now corrected this typo.
>
> What we intended to convey is that LoRA requires $(m+n)(b-1)r$ additional parameters to increase the rank from $r$ to $br$, whereas BoRA only requires $b^2r$ parameters, which is much more efficient.
>
> In practice, the number of trainable parameters was obtained by directly counting all the trainable parameters in the network, rather than calculating them from the formula. Therefore, this typo does not affect the overall correctness of the experimental comparisons or any other parts of the paper.
>
> `Due to space limitations, please refer to Part II for our responses to other questions.`

---

> ### Author Response · Authors · 2025-11-21
> **Response to Reviewer 9WMS [Part II]**
>
> ***`Q3: "theoretical justification that BoRA can achieve the rank bound under certain assumptions"`***
>
> **R3:**
> In Section 3.3, we briefly discussed the theoretical justification for Proposition 1. Here, we provide a more detailed expansion and explain under which assumptions BoRA’s rank can achieve the upper bound presented in Proposition 1.
>
> For a pre-trained weight $W_0 \in \mathbb{R}^{m \times n}$, LoRA maintains two low-rank matrices $B \in \mathbb{R}^{m \times r}$ and $A \in \mathbb{R}^{r \times n}$. Let matrix $B \in \mathbb{R}^{m \times r} = [B_1, B_2, \dots, B_b]^\top$ be divided into $b$ blocks, where $B_i \in \mathbb{R}^{\frac{m}{b} \times r}$. Similarly, matrix $A \in \mathbb{R}^{r \times n} = [A_1, A_2, \dots, A_b]$ is divided into $b$ blocks, where $A_i \in \mathbb{R}^{r \times \frac{n}{b}}$.
>
> The LoRA weights generated by the proposed BoRA can be denoted as follows:
> $$
> \Delta W =
> \left [
> \begin{array}{ccccc}
> B_1 \Sigma_{1,1} A_1 &  B_1 \Sigma_{1,2} A_2 & \dots & B_1 \Sigma_{1,b} A_b \\\\
> B_2 \Sigma_{2,1} A_1 &  B_2 \Sigma_{2,2} A_2 & \dots & B_2 \Sigma_{2,b} A_b \\\\
> \vdots & \vdots & \vdots & \vdots \\\\
> B_b \Sigma_{b,1} A_1 &  B_b \Sigma_{b,2} A_2 & \dots & B_b \Sigma_{b,b} A_b \\\\
> \end{array}
> \right ]
> $$
> where $\Sigma_{i,j} \in \mathbb{R}^{r \times r} \ (\forall i,j \in [b])$ are diagonal matrices.
>
> Note that $\Delta W$ can also be represented as the product of three matrices ($B' \in \mathbb{R}^{m \times br}$, $\Sigma' \in \mathbb{R}^{br \times br}$, and $A' \in \mathbb{R}^{br \times n}$), as shown below:
> $$
> \Delta W = B'\Sigma'A' =
> \left [
> \begin{array}{ccccc}
> B_1 & 0 & \dots & 0 \\\\
> 0 & B_2 & \dots & 0 \\\\
> \vdots & \vdots & \vdots & \vdots \\\\
> 0 & 0 & \dots & B_b \\\\
> \end{array}
> \right ]
> \left [
> \begin{array}{ccccc}
> \Sigma_{1,1}& \Sigma_{1,2}& \dots &\Sigma_{1,b}\\\\
> \Sigma_{2,1}& \Sigma_{2,2}& \dots &\Sigma_{2,b}\\\\
> \vdots & \vdots & \vdots & \vdots \\\\
> \Sigma_{b,1}& \Sigma_{b,2}& \dots &\Sigma_{b,b}\\\\
> \end{array}
> \right ]
> \left [
> \begin{array}{ccccc}
> A_1 & 0 & \dots & 0 \\\\
> 0 & A_2 & \dots & 0 \\\\
> \vdots & \vdots & \vdots & \vdots \\\\
> 0 & 0 & \dots & A_b \\\\
> \end{array}
> \right ]
> $$
> Based on the shapes of the matrices, we can derive the upper bound for the rank of the three matrices:
> $$
> \text{rank}(B')\leq \min\{m,br\},\
> \text{rank}(A')\leq \min\{n,br\},\
> \text{rank}(\Sigma')\leq br
> $$
> According to the properties of matrix multiplication:
> $$
> \text{rank}(\Delta W)\leq \min\{\text{rank}(A'),\text{rank}(\Sigma'),\text{rank}(B')\} = \min\{m,n,br\}.
> $$
> It is important to note that $\text{rank}(\Delta W)$ can only have a clearly defined upper bound, with the minimum value being 0. For example, in BoRA, the matrix $B$ is initialized to zero, so at the beginning of training, $\text{rank}(B') = 0$, which results in $\text{rank}(\Delta W) = 0$.
>
> However, when all three matrices are full rank (i.e., $\text{rank}(B') = \min\{m, br\}$, $\text{rank}(A') = \min\{n, br\}$, and $\text{rank}(\Sigma') = br$), equality holds, and the upper bound for $\text{rank}(\Delta W)$ can be achieved. This condition is typically satisfied because $\Sigma'$ and $A'$ are initialized randomly, which usually ensures that $\text{rank}(A') = \min\{n, br\}$ and $\text{rank}(\Sigma') = br$. As training progresses, $B'$ is continually updated, and $\text{rank}(B')$ can gradually approach full rank. This is evident from Figure 4 in the paper, where the effective rank of BoRA (the number of singular values greater than 0.005) approaches $br$.
>
> We will add explanations and theoretical justifications for Proposition 1 in both the main text and the appendix.
>
>
> ***`Q4 "why is 0.005 chosen as the threshold of singular values"`***
>
> **R4:** The threshold is introduced to calculate the effective rank.
> For a matrix $M \in \mathbb{R}^{m \times n}$, without loss of generality, assume $n \leq m$. The standard rank of $M$ is defined as:
> $$\text{rank}(M)=\max\{i∣\sigma_i>0\}\leq n$$
> where $\sigma_1\geq \sigma_2\geq \dots \geq \sigma_n$ are the singular values of $M$.
> The concept of effective rank aims to treat singular values close to zero as zero. A simple definition of effective rank is:
> $$\text{erank}(M)=\max\{i∣\sigma_i>\epsilon\}\leq \text{rank}(M)$$
> where $\epsilon$ is the threshold for calculating the effective rank.
>
> In this paper, we set the threshold $\epsilon = 0.005$, following the HiRA (Huang et al., 2025) method (see Figure 4 in [https://openreview.net/forum?id=TwJrTz9cRS]). We will include additional explanations in the paper.
>
> **`Lastly, thank you again for your valuable feedback, which has greatly improved our work. We hope our responses address your concerns and clarify any doubts. Please feel free to reach out to us if you have any further questions.`**

---

> ### Comment · Reviewer_9WMS · 2025-11-21
>
> Thanks to the authors for their reply. My concerns have all been addressed, so I raise my rating to 6.

---

> > ### Author Response · Authors · 2025-11-22
> > **Sincere Thanks for Your Support**
> >
> > Thank you once again for reviewing our paper and for your thoughtful feedback. We really appreciate your recognition of our response and the decision to raise the score. Your comments have been very helpful in improving the quality of the work.

---

### Official Review · Reviewer_YooA · 2025-10-30

**Soundness:** 2
**Presentation:** 2
**Contribution:** 2
**Rating:** 6
**Confidence:** 4

**Summary:**

This paper proposes a parameter-efficient fine-tuning method called **Block Diversified Low-Rank Adaptation (BoRA)**, which improves the rank of LoRA weights through matrix block  diversification. The core idea of BoRA is to divide the LoRA weight matrices $A$ and $B$ into $b$ blocks the columns and rows. To break the correlation between the block products $B_iA_j$ in different rows or columns, BoRA introduces a unique diagonal matrix,  $\Sigma_{i,j}$ , for each block multiplication, resulting in $B_i\Sigma_{i,j}A_j$. This formulation increases the upper bound of the LoRA weight rank by $b$ times while adding only $b^2r$ additional parameters. To demonstrate the effectiveness of BoRA, the authors conduct extensive experiments on natural language understanding, mathematical reasoning, and commonsense reasoning tasks, covering both RoBERTa and various mainstream LLMs. The results show that, compared with LoRA and three of its variants, BoRA consistently achieves stable performance improvements. Under comparable parameter settings, BoRA improves accuracy by about 2-4\% over LoRA.

**Strengths:**

- The paper analyzes the rank limitation of standard LoRA from the perspective of block matrix multiplication, showing that the correlation between block matrices constrains the expressiveness of LoRA and thus provides a strong theoretical foundation for the proposed solution.
- It introduces a LoRA variant called BoRA, which eliminates the correlation between block matrices by introducing diagonal matrices, achieving substantial performance improvement with only a small parameter overhead of $b^2r$.
- The authors also prove that LoRA and MELoRA are special cases of BoRA, thereby establishing a clear connection with previous work.
- Comprehensive evaluations on models with different architectures and across various task types demonstrate the broad applicability of the BoRA.

**Weaknesses:**

- Although the authors mention inference latency in the appendix, they lack efficiency comparisons during training (e.g., convergence time, memory usage, training latency), and adding such comparisons would make the work more convincing.
- The authors claim that BoRA raises the rank upper bound, and in Related Work they also mention HiRA and KronA as methods that increase the rank of LoRA weights, yet these are not included as baselines for comparison.
- In the original MELoRA paper, the number of epochs for GLUE tasks is much larger (e.g., 60 for SST-2), whereas this paper uses relatively small epoch counts (e.g., 2 for SST-2). This discrepancy may affect the resulting performance.
- When discussing different tuning granularity, the authors compare BoRA only with LoRA and do not compare it with other baselines.

**Questions:**

1. Could you provide efficiency comparisons during training (e.g., convergence time, memory usage, training latency)?
2. When dividing the matrices into $b$ blocks, on matrices of what size is this partitioning performed?
3. Why are the ranks of the other baselines set to 8, while HydraLoRA is set to 4? Could you also provide results for HydraLoRA with rank 8?
4. Why did the authors choose relatively small numbers of training epochs?
5. In Fig. 3(b), standard LoRA improves as $r$ increases, whereas BoRA with $b=16$ or $64$ shows performance drops as $r$ increases (e.g., for $b=16, r=8 \ vs. r=16$). Could you explain the reason for this behavior?
6. For Different Tuning Granularity, why are comparisons made only between BoRA and LoRA, without including other baselines?
7. Is the notation $i, j \in [r]$ in Section 3.2 a typo by the authors? Should $r$ be $b$ here?

---

> ### Author Response · Authors · 2025-11-21
> **Response to Reviewer YooA [Part I]**
>
> **Hi, Reviewer YooA:**
>
> Thank you for acknowledging the **strong theoretical foundation** of our method and its **clear connection with previous work**, as well as the the **substantial performance improvement** and the **broad applicability** of BoRA.
>
> We note that your primary concerns relate to several aspects of the experimental evaluation, including **training-efficiency comparisons**, **comparisons with HiRA and KronA**, the **number of training epochs used for GLUE tasks**, and **the performance of other baselines under different tuning granularity**. Below, we address each of these points in detail.
>
>
> ***`Q1: "training-efficiency comparisons"`***
>
> **R1:** As suggested, we measured the wall-clock time and memory usage of various methods. One epoch of training on the mathematical reasoning task was performed using the LLama-3-8B model, with an Nvidia A6000 GPU and a batch size of 4.
>
> | | # Param (M) | Memory (GB) | Time (Min)  |
> |---|---|---|---|
> | LoRA($r=8$)      | 4.72 | 23.57 | 18.5 |
> | BoRA($r=8,b=16$) | 4.92 | 23.63 | 18.6 |
> | MELoRA($r=8$)    | 4.72 | 23.57 | 18.6 |
> | HydraLoRA($r=4$) | 5.11 | 24.98 | 19.6 |
> | DoRA($r=8$)      | 4.92 | 24.36 | 22.3 |
>
> The results show the following:
>
> * **BoRA** and **MELoRA** exhibit nearly identical training time and memory footprint compared to LoRA.
> * **HydraLoRA** introduces additional gating layers, which add minor computational cost and result in a small increase in runtime.
> * **DoRA**, due to its reparameterization of directions and magnitudes, requires more complex computation and therefore introduces some overhead as well.
>
> Overall, the empirical evidence supports our argument that the end-to-end training overhead of BoRA is essentially **on par with LoRA, and comparable or lower than several other state-of-the-art methods**. These results will be included in the paper and uploaded in the next few days.
>
> ***`Q2: "comparison with HiRA and KronA"`***
>
> **R2:**
> As suggested, we have added comparisons with **HiRA** (Huang et al., 2025) and **KronA** (Edalati et al., 2023).
>
> | LLama-3-8B | #Params | AddSub | MultiArith | SingleEq | gsm8k | AQuA  | SVAMP | Avg    |
> |--------------------|---------|--------|------------|----------|-------|-------|-------|--------|
> | LoRA($r=8$)| 4.72M   | 82.28  | 87.06      | 91.60    | 55.65 | 24.02 | 68.53 | 68.19  |
> | HiRA($r=8$)| 4.72M   | 82.93  | 89.65      | 92.40    | 57.47 | 26.51 | 69.87 | 69.81  |
> | BoRA($r=8,b=16$)   | 4.92M   | 88.35  | 93.00      | 92.72    | 58.83 | 27.17 | 73.40 | 72.24  |
> | KronA      | 0.52M   | 77.97  | 84.33      | 87.40    | 48.82 | 25.59 | 62.70 | 64.47  |
> | BoRA+($r=8,b=16$)  | 0.20M   | 84.56  | 88.67      | 90.94    | 53.48 | 25.77 | 67.10 | 68.42  |
> | BoRA+($r=16,b=16$) | 0.39M   | 85.06  | 90.17      | 91.34    | 53.31 | 25.44 | 67.60 | 68.82  |
>
>
> * **HiRA** increases the rank of LoRA weights by employing the Hadamard product with the formulation $\Delta W = W_0 \odot (BA)$, where $W_0$ is the pre-trained weight. As shown above, HiRA indeed achieves slightly better accuracy than LoRA; however, its performance still falls noticeably short compared to BoRA under a similar parameter budget.
>
> * Similarly, **KronA** increases the rank of LoRA weights by employing the Kronecker product with the formulation $\Delta W = B \otimes A \in \mathbb{R}^{m \times n}$. Here, $B \in \mathbb{R}^{b_1 \times b_2}$, $A \in \mathbb{R}^{a_1 \times a_2}$, and $a_1 b_1 = m$, $a_2 b_2 = n$. Following the original paper, we set $a_1$ and $b_1$, as well as $a_2$ and $b_2$, to be as close as possible to minimize the number of parameters. As a result, KronA has significantly fewer parameters than LoRA and BoRA. To compare fairly with KronA and inspired by **Reviewer dURX**, we constructed a variant of BoRA by freezing matrices $A$ and $B$, applying PiSSA (Meng et al., 2024) to initialize them (via SVD on the pre-trained weight), and initializing the learnable block-wise diagonal parameters $\sigma$ to zero. We denote this variant as **BoRA+**. As shown above, BoRA+ matches KronA in parameter count but achieves substantially higher accuracy.
>
> `Due to space limitations, please refer to Part II for our responses to other questions.`

---

> ### Author Response · Authors · 2025-11-21
> **Response to Reviewer YooA [Part II]**
>
> ***`Q3: "number of training epochs used for GLUE tasks"`***
>
> **R3:**
> The decision to use fewer epochs was based on the observation that the model converges quickly with a small number of epochs, and further training results in only marginal improvements. For example, in our experiments, LoRA ($r=8$) achieves an accuracy of 93.46% on SST-2 after just 2 epochs, while in the MELoRA paper, LoRA ($r=8$) reaches only 94.50% after 60 epochs.
>
> Therefore, to ensure efficiency in our experiments, we used a smaller number of epochs. However, we carefully tuned LoRA's learning rate to ensure optimal performance, and BoRA directly used the same learning rate as LoRA. This ensures a fair comparison between the two methods. A similar approach of using fewer epochs for GLUE tasks has been adopted in recent works.
> For example, both ReLoRA [1] and LoRA-GA [2]  reduced the training epochs for GLUE tasks to just one.
>
> To further ensure rigor, we trained both LoRA and BoRA for 30 epochs on the SST-2 dataset. As shown below, even after 30 epochs, BoRA continues to outperform LoRA. Additionally, we observed that LoRA's performance improvement plateaued early, with little to no increase after the first few epochs. We will provide a more detailed explanation for using fewer epochs in the paper and include the full results for longer training periods in the appendix.
>
> | RoBERTa-Base on SST-2 | 2 epochs | 30 epochs  |
> |-----------------------|----------|------------|
> | LoRA($r=8$)   | 93.46    | 94.51      |
> | BoRA($r=8,b=8$)       | 93.92    | 94.89      |
>
> [1] ReLoRA: High-Rank Training Through Low-Rank Updates. ICLR 2024. \
> [2] LoRA-GA: Low-Rank Adaptation with Gradient Approximation. NeruIPS 2024.
>
>
>
> ***`Q4: "the performance of other baselines under different tuning granularity"`***
>
> **R4:**
> The purpose of the experiment on different tuning granularity is to demonstrate BoRA's scalability across different layers, showing that BoRA can provide performance improvements at each layer. As a result, we primarily compared the performance of BoRA with LoRA. However, we have now included the experimental results for other baseline methods under different tuning granularity, based on the results presented in Figure 3(c). As shown below, BoRA still significantly outperforms these baselines in any given tuning granularity.
>
> |  | Q | QV| QKV       | QKVUD     | QKVUDOG    |
> |------------------|-----------|-----------|-----------|-----------|------------|
> | LoRA($r=8$)      | 64.97     | 68.05     | 68.19     | 72.14     | 72.23      |
> | DoRA($r=8$)      | 65.52     | 68.62     | 68.41     | 72.54     | 72.66      |
> | MELoRA($r=8$)    | 65.30     | 68.22     | 68.14     | 72.36     | 72.64      |
> | HydraLoRA($r=4$) | 65.01     | 67.23     | 67.58     | 72.03     | 72.22      |
> | BoRA($r=8,b=16$) | **67.87** | **71.01** | **72.24** | **73.40** | **73.88**  |
>
>
>
>
> ***`Q5: "on matrices of what size is this partitioning (into $b$ blocks) performed"`***
>
> **R5:** The partitioning into $b$ blocks is performed on the LoRA matrices $A$ and $B$. Specifically, for a pre-trained weight $W_0 \in \mathbb{R}^{m \times n}$, LoRA maintains two low-rank matrices: $B \in \mathbb{R}^{m \times r}$ and $A \in \mathbb{R}^{r \times n}$. Matrix $B = [B_1, B_2, \dots, B_b]^\top$ is divided into $b$ blocks, where each block $B_i \in \mathbb{R}^{\frac{m}{b} \times r}$. Similarly, matrix $A = [A_1, A_2, \dots, A_b]$ is divided into $b$ blocks, where each block $A_i \in \mathbb{R}^{r \times \frac{n}{b}}$.
>
> For example, in the case of LLama-3-8B, the values of $m$ and $n$ are typically 1024 or 4096, while $r$ and $b$ are usually set to values such as 8 or 16 in our experiments.
>
> `Due to space limitations, please refer to Part III for our responses to other questions.`

---

> ### Author Response · Authors · 2025-11-21
> **Response to Reviewer YooA [Part III]**
>
> ***`Q6: "why HydraLoRA uses rank 4 instead of rank 8"`***
>
> **R6:**
> The use of rank 4 in HydraLoRA ensures that its parameter size is comparable to that of LoRA and other methods. HydraLoRA follows a Mixture of Experts (MoE) structure, using a single $A$ matrix and multiple $B$ matrices, which are weighted by a gating network to produce the final $B$ matrix. In our experiments, we used 3 $B$ matrices, and HydraLoRA ($r=4$) has a parameter size roughly equivalent to that of LoRA ($r=8$). Due to the gating mechanism, HydraLoRA’s parameter count may even be slightly higher.
>
> We also provide results for HydraLoRA with $r=8$ using LLama-3-8B. As shown below, HydraLoRA ($r=8$) has a significantly larger parameter size than BoRA ($r=8, b=16$), but its accuracy is much lower.
>
> | LLama-3-8B| #Params | AddSub | MultiArith | SingleEq | gsm8k | AQuA  | SVAMP | Avg    |
> |-------------------|---------|--------|------------|----------|-------|-------|-------|--------|
> | LoRA($r=8$)       | 4.72M   | 82.28  | 87.06      | 91.60    | 55.65 | 24.02 | 68.53 | 68.19  |
> | HydraLoRA($r=4$)  | 5.11M   | 77.22  | 89.17      | 91.73    | 56.63 | 24.41 | 66.30 | 67.58  |
> | HydraLoRA($r=8$)  | 9.04M   | 86.08  | 91.00      | 91.14    | 55.50 | 25.98 | 68.10 | 69.63  |
> | BoRA($r=8, b=16$) | 4.92M   | 88.35  | 93.00      | 92.72    | 58.83 | 27.17 | 73.40 | 72.24  |
>
> | LLama-3-8B| #Params | boolq | piqa  | social_i_qa | hellaswag | winogrande | ARC-Challenge | ARC-Easy | openbookqa | Avg    |
> |-------------------|---------|-------|-------|-------------|-----------|------------|---------------|----------|------------|--------|
> | LoRA($r=8$)       | 4.72M   | 73.17 | 89.34 | 80.64       | 93.22     | 87.42      | 80.20 | 92.51    | 87.13      | 85.45  |
> | HydraLoRA($r=4$)  | 5.11M   | 72.66 | 88.85 | 81.01       | 93.48     | 87.37      | 80.20 | 92.68    | 87.80      | 85.51  |
> | HydraLoRA($r=8$)  | 9.04M   | 72.66 | 89.39 | 80.96       | 93.84     | 87.53      | 81.14 | 92.85    | 88.20      | 85.82  |
> | BoRA($r=8, b=16$) | 4.92M   | 73.88 | 89.17 | 81.99       | 94.11     | 88.00      | 82.00 | 93.14    | 88.60      | 86.36  |
>
>
> ***`Q7: "why BoRA ($b=16$ and $b=64$) shows performance drops as rank increases in Fig. 3(b)"`***
>
> **R7:**
> The slight performance drops observed in BoRA as rank increases can be attributed to the **inherent variability in experimental results**. Specifically, Fig. 3(b) shows the average accuracy for LLama-3-8B on mathematical reasoning tasks. As shown in Table 5 in the appendix, the **standard deviation of BoRA** using LLama-3-8B on the mathematical reasoning task ranges **from 0.38 to 0.61**.
>
> In Fig. 3(b), when $b=16$, the accuracy drops by 0.23 as the rank increases from $r=8$ to $r=16$. This drop is within the expected fluctuation range and is negligible compared to the improvement BoRA shows over LoRA. Additionally, when $b=8$ and $b=32$, the accuracy increases by only 0.12 and 0.26, respectively, as the rank increases from $r=8$ to $r=16$.
>
> These results further suggest that increasing the rank (i.e., $r$) leads to less significant performance improvements than increasing the number of blocks (i.e., $b$) in BoRA.
>
>
> ***`Q8: "typo in Line 178, where $i,j \in [r]$ should be $i,j \in [b]$""`***
>
> **R8:** Thank you for your thorough review.
> We have corrected the typo in **Line 178** (from $i,j \in [r]$ to $i,j \in [b]$), and the updated PDF will be uploaded in the next few days.
>
>
> **`Lastly, thank you again for your valuable feedback, which has greatly improved our work. We hope our responses address your concerns and clarify any doubts. Please feel free to reach out to us if you have any further questions.`**

---

> ### Comment · Reviewer_YooA · 2025-11-26
> **Thank you for the response.**
>
> Thanks to the authors for their reply.
>
> ### Choose a small number of training epochs
>
> If the authors consider a 1% improvement to be marginal, then the proposed method in this paper seems equally trivial in most situations. Additionally, the experiments suggest that both the baselines and BoRA may not be fully trained, as shown by the table in your response to Q3. The authors should report results for 60 epochs rather than 30 in this table. A limited training period (2 epochs) fails to capture the real performance of the models, raising concerns about the reliability and credibility of the results. Therefore, **the experimental findings in this paper are questionable and lack sufficient rigor**.
>
> ### Weak Scalability
>
> In Fig. 3(b), standard LoRA improves as \( r \) increases, whereas BoRA with \( b = 16 \) or 64 shows performance drops as \( r \) increases. This instability in BoRA’s performance limits its practical applicability since the BoRA mechanism is an improvement over LoRA.
>
> After reviewing the rebuttal, the primary issue with this work is that the introduction of the BoRA mechanism as an improvement over LoRA fails to demonstrate a clear advantage, largely due to the unfair experimental setup.  **This raises significant concerns about the reliability and credibility of the results.** Thanks to the authors for their reply.

---

> > ### Author Response · Authors · 2025-11-28
> > **Second Response to Reviewer YooA [Part I]**
> >
> > Dear Reviewer YooA,
> >
> > Thank you for carefully reading our previous response and for providing further constructive feedback. In response to the two remaining concerns, we have conducted additional experiments and provide detailed explanations below.
> >
> > ***`Concern 1: "choose a small number of training epochs"`***
> >
> > **R1:**
> > Thank you for your valuable comment. It helped us realize that our earlier statement — “further training results in only marginal improvements” — was not sufficiently precise. In particular, for RoBERTa-Base on GLUE, a 1% improvement is certainly not marginal. As shown in Table 1 of our paper, even increasing the rank by a factor of four ($r$ = 8 to $r$ = 32) results in only about a 1% improvement (from 82.55% to 83.60%).
> >
> > Our intended point was that if achieving such a 1% gain requires increasing the training cost by more than 30 times (for example, training for 60 epochs instead of 2), then that improvement may not be cost-effective. In contrast, BoRA achieves a 1% improvement for RoBERTa-Base on GLUE under the same training budget (the same number of epochs) and with a similar number of parameters, which makes this gain meaningful. For efficiency, we used fewer epochs on the GLUE benchmark, but we always ensured that BoRA, LoRA, and all baselines used exactly the same number of epochs to maintain fair comparisons.
> >
> > Your comment correctly pointed out, however, that using relatively small epoch numbers may reduce the rigor of the evaluation. To address this concern and further verify that BoRA still outperforms baselines when all methods are sufficiently trained, we conducted additional experiments using substantially more training epochs.
> > Specifically, we strictly followed the epoch settings used in MELoRA and repeated the RoBERTa-Base experiments accordingly. The numbers of epochs for RTE, CoLA, and SST-2 were 80, 80, and 60, respectively. The results are shown below and are consistent with those reported in Table 4 of MELoRA [https://arxiv.org/abs/2402.17263]. These additional experiments demonstrate that even under extensive training, the accuracy improvements provided by BoRA remain valid. We will include results from longer epoch training in our paper to strengthen the rigor of our GLUE evaluation.
> >
> >
> > |  | #Params | RTE | CoLA | SST-2 | Avg.  |
> > |---|---|---|---|---|---|
> > | LoRA($r=8$) | 0.29M | 75.81 | 62.07 | 94.62 | 77.50  |
> > | MELoRA($r=8$) | 0.29M | 76.43 | 62.97 | 94.35 | 77.92  |
> > | BoRA($r=8, b=8$) | 0.31M | 76.98 | 63.93 | 95.34 | 78.75  |

---

> > ### Author Response · Authors · 2025-11-28
> > **Second Response to Reviewer YooA [Part II]**
> >
> > ***`Concern 2: "weak scalability"`***
> >
> > **R2:**
> > First, we would like to provide additional information regarding the accuracy changes of LoRA at larger ranks. As shown in the table below, when the rank ($r$) is small, increasing $r$ leads to a significant improvement in performance. However, as $r$ becomes larger, the improvement from further increasing $r$ becomes relatively small. For instance, the accuracy improvement was only 0.1-0.3% when the rank increased from 128 to 256, or from 256 to 512. This is primarily because once the effective rank of the LoRA weights reaches a certain threshold, the model's expressiveness tends to saturate, and further increases in rank have limited impact on improving accuracy. In such cases, if the experimental variability exceeds the magnitude of this improvement, a single experiment might even show a slight decrease in performance as the rank increases.
> >
> > | Llama-3-8B/Mathematical | Accuracy | Relative Improvement vs. Half Rank  |
> > |---|---|---|
> > | LoRA($r=2$) | 65.70 | -  |
> > | LoRA($r=4$) | 66.57 | 0.87   |
> > | LoRA($r=8$) | 68.19 | 1.62   |
> > | LoRA($r=16$) | 70.19 | 2.00   |
> > | LoRA($r=32$) | 71.67 | 1.48   |
> > | LoRA($r=64$) | 72.41 | 0.74   |
> > | LoRA($r=128$) | 72.98 | 0.57   |
> > | LoRA($r=256$) | 73.06 | **0.08**   |
> > | LoRA($r=512$) | 73.43 | **0.37**   |
> >
> > The instability in BoRA's performance that you mentioned is essentially due to this phenomenon. With smaller values of $b$ (e.g., 2 or 4), increasing $r$ results in a significant accuracy improvement because the model's expressive capacity has not yet saturated. However, for larger values of $b$ (e.g., $b$ = 16 or 64), BoRA generally achieves a higher effective rank, and further increases in $r$ do not lead to as significant an accuracy improvement. Nevertheless, BoRA consistently outperforms LoRA, regardless of the value of r, even when $b$ is smaller (e.g., 4, 8, 16).
> >
> > Finally, it should be noted that the initial ablation experiments (Figure 3(b)) were based on a single trial and were not repeated. We now repeat the experiments for $b$ = 16 and 64, each conducted three times. For $b$ = 16, the range of $r$ is [8, 16, 32]; for $b$ = 64, the range of $r$ is [16, 32, 64]. The results presented below demonstrate that, by repeating the experiments and reducing inherent fluctuations, BoRA’s accuracy gradually improves with increasing $r$ for both $b$ = 16 and $b$ = 64, although the improvement is relatively small. This is because the model's expressive power tends to saturate, which is not a limitation of BoRA.
> >
> > | Llama-3-8B/Mathematical | AddSub | MultiArith | SingleEq | gsm8k | AQuA | SVAMP | Avg  |
> > |---|---|---|---|---|---|---|---|
> > | BoRA($r=8, b=16$) | 86.84 | 94.67 | 93.70 | 60.65 | 25.20 | 71.70 | 72.13  |
> > | BoRA($r=16, b=16$) | 88.36 | 94.17 | 93.21 | 59.82 | 23.82 | 74.65 | 72.34  |
> > | BoRA($r=32, b=16$) | 89.87 | 93.83 | 93.11 | 59.82 | 25.20 | 75.80 | 72.94  |
> > |  |  |  |  |  |  |  |   |
> > | BoRA($r=16, b=64$) | 89.62 | 95.67 | 93.31 | 60.50 | 23.62 | 75.70 | 73.07  |
> > | BoRA($r=32, b=64$) | 89.03 | 95.44 | 93.83 | 59.72 | 28.61 | 73.57 | 73.37  |
> > | BoRA($r=64, b=64$) | 90.89 | 95.17 | 93.90 | 59.29 | 30.31 | 74.20 | 73.96  |
> >
> > **`Once again, thank you for your valuable feedback, and please let us know if you have any further questions or suggestions.`**

---

### Official Review · Reviewer_dURX · 2025-11-01

**Soundness:** 3
**Presentation:** 3
**Contribution:** 3
**Rating:** 6
**Confidence:** 4

**Summary:**

This paper proposes BoRA, which addresses the rank-constrained problem of LoRA by enhancing diversity through matrix partitioning and the introduction of a block diagonal matrix, thereby increasing the weight rank of LoRA by a factor of b with only a few additional parameters. Multi-model, multi-task experiments show that it outperforms LoRA and its variants, achieving a 2-4% accuracy improvement with similar parameters.

**Strengths:**

- The methodology is clearly and comprehensively described, and Figure 1 clearly explains BoRA and its differences from other methods.

- The experiments in the paper are comprehensive, validating the effectiveness of the method on different tasks and models.

**Weaknesses:**

- The "Results of Different Tuning Granularity" section in Section 4.4 does not guarantee that the number of parameters or computational costs will be consistent (or as consistent as possible).

- Although the authors analyzed the efficiency of BoRA, experiments comparing the efficiency of different methods are lacking.

**Questions:**

- Why must the diagonal matrix $\Sigma $ be non-negative?

- Is it feasible to freeze the existing trained LoRA structure and train only the diagonal matrix $\Sigma $?

- Will the matrix rank change with different layers and training periods?

---

> ### Author Response · Authors · 2025-11-21
> **Response to Reviewer dURX [Part I]**
>
> **Hi, Reviewer dURX:**
>
> Thank you for acknowledging the **clarity** of our presentation, the **comprehensiveness** of our experiments, and the **effectiveness** of BoRA.
>
> We note that your primary concerns focus on the **experimental comparison of training efficiency**. Additionally, there may be a **misunderstanding regarding the purpose of the tuning-granularity experiment** in Section 4.4 (i.e., Figure 3(c)). Below, we address each of these points in detail.
>
>
> ***`Q1: "experiments of different tuning granularity dose not guarantee consistent parameters or computational costs"`***
>
> **R1:** Sorry for the confusion, but we believe there is a misunderstanding regarding the purpose of the tuning-granularity experiment.
>
> To clarify, within any given tuning granularity, the number of parameters and computational costs in BoRA are approximately the same as in LoRA. The goal of this experiment is **to compare BoRA’s performance relative to LoRA at each granularity**, rather than comparing BoRA’s performance across different granularities.
>
> Figure 3(c) successfully demonstrates that BoRA’s performance improvement over LoRA is consistent across various layers of the model (e.g., k_proj, q_proj).
>
>
> ***`Q2: "experimental comparison of training efficiency"`***
>
> **R2:** As suggested, we measured the wall-clock time and memory usage of various methods. One epoch of training on the mathematical reasoning task was performed using the LLama-3-8B model, with an Nvidia A6000 GPU and a batch size of 4.
>
> | | # Param (M) | Memory (GB) | Time (Min)  |
> |---|---|---|---|
> | LoRA($r=8$)      | 4.72 | 23.57 | 18.5 |
> | BoRA($r=8,b=16$) | 4.92 | 23.63 | 18.6 |
> | MELoRA($r=8$)    | 4.72 | 23.57 | 18.6 |
> | HydraLoRA($r=4$) | 5.11 | 24.98 | 19.6 |
> | DoRA($r=8$)      | 4.92 | 24.36 | 22.3 |
>
> The results show the following:
>
> * **BoRA** and **MELoRA** exhibit nearly identical training time and memory footprint compared to LoRA.
> * **HydraLoRA** introduces additional gating layers, which add minor computational cost and result in a small increase in runtime.
> * **DoRA**, due to its reparameterization of directions and magnitudes, requires more complex computation and therefore introduces some overhead as well.
>
> Overall, the empirical evidence supports our argument that the end-to-end training overhead of BoRA is essentially **on par with LoRA, and comparable or lower than several other state-of-the-art methods**. These results will be included in the paper and uploaded in the next few days.
>
>
> ***`Q3: "why must the diagonal matrix be non-negative"`***
>
> **R3:**
> This confusion likely arises from our statement in Line 194: "To prevent information loss associated with a zero value in $\Sigma$, we further apply the exponential function to $\sigma$, ensuring that $\Sigma$ contains only positive values."
> In fact, the diagonal matrix $\Sigma_{i,j}$ is required to be **non-zero, rather than non-negative**. A more detailed explanation is provided below.
>
> First, $\Sigma_{i,j}$ is formulated as follows:
> $$\Sigma_{i,j} = \text{Diag}(\text{Exp}(\text{Norm}(\sigma_{i,j})))$$
> where $\sigma_{i,j}$ are learnable parameters, initialized in the same way as matrix $A$ (i.e., zero-centered using Kaiming_Uniform), with the same initialization scale. This initialization ensures that $\sigma_{i,j}$ can be efficiently optimized with the same learning rate as $A$.
>
> However, the optimization of $\Sigma_{i,j}$ faces two main challenges:
> 1. $\sigma_{i,j}$ is initialized with a small variance, causing the differences between $\sigma_{i,j}$ across blocks $(i, j)$ to be very small, which fails to increase the diversity between blocks. To address this, we apply the Norm operation on $\sigma_{i,j}$ to mitigate the effect of the small initialization.
>
> 2. $\text{Norm}(\sigma_{i,j})$ remains zero-centered, and zero or near-zero values can nullify the values in $B_i$ and $A_j$. To solve this, we apply the exponential function, transforming the zero-centered values into one-centered values.
>
> In fact, the second problem could also be addressed by directly setting:
> $$\Sigma_{i,j} = \text{Diag}(\text{Addone}(\text{Norm}(\sigma_{i,j}))) = \text{Diag}(1+\text{Norm}(\sigma_{i,j}))$$
> However, this "Addone" strategy can still result in some values close to zero in $\Sigma_{i,j}$.
> Experimental comparisons on mathematical reasoning tasks show that while adding one does improve accuracy, the exponential function leads to better performance.
>
> | BoRA ($r=8, b=16$) | Gemma-7B | LLama-3-8B | Qwen2.5-14B    |
> |---|---|---|---|
> | w. Norm($\cdot$)         | 71.77 | 68.51 | 79.79 |
> | w. Exp(Norm($\cdot$))    | **73.10** | **72.24** | **80.60** |
> | w. Addone(Norm($\cdot$)) | 72.53 | 71.38 | 80.14 |
>
> We will update the relevant statements in Section 3.2 of the paper accordingly.
>
> `Due to space limitations, please refer to Part II for our responses to other questions.`

---

> ### Author Response · Authors · 2025-11-21
> **Response to Reviewer dURX [Part II]**
>
> ***`Q4: "is it feasible to freeze the existing trained LoRA structure and train only the diagonal matrix"`***
>
> **R4:**
> Thank you for this insightful question! It has inspired some great ideas.
>
> Freezing matrices $A$ and $B$ is definitely possible, but it **requires some adjustments**. In both LoRA and BoRA, matrix $B$ is initialized to zero. Freezing $B$ directly would cause the change in the pre-trained weights (i.e., $\Delta W$) to remain zero. To address this, we draw inspiration from PiSSA (Meng et al., 2024) and initialize both $A$ and $B$ with non-zero values. Specifically, we apply Singular Value Decomposition (SVD) to the pre-trained weights, extracting the first $r$ singular values and their corresponding singular vectors to initialize matrices $A$ and $B$. The remaining singular values and vectors are stored in the pre-trained weights. Finally, we initialize the learnable block diagonal parameter $\sigma_{i,j}$ to zero so that $\Sigma_{i,j}$ is initialized as the identity matrix. This prevents introducing noise into the first $r$ singular values and vectors before training.
>
> This variant, which we call **BoRA+**, is **a lighter version of BoRA**.
> In BoRA+, matrices $A$ and $B$ are frozen, and only the block-wise diagonal matrices $\Sigma_{i,j}$ are optimized.
> As shown below, we report the accuracy of LLama-3-8B on the mathematical reasoning task using different methods.
> PiSSA ($r=8$) achieves slightly higher accuracy than LoRA ($r=8$). However, BoRA+ ($r=8, b=16$), initialized with PiSSA's method, requires fewer trainable parameters to match PiSSA's accuracy. Increasing $b$ in BoRA+ to match the number of trainable parameters in PiSSA further improves the accuracy, surpassing PiSSA and significantly outperforming LoRA.
>
>
> | LLama-3-8B        | #Params | AddSub | MultiArith | SingleEq | gsm8k | AQuA  | SVAMP | Avg    |
> |-------------------|---------|--------|------------|----------|-------|-------|-------|--------|
> | LoRA($r=8$)       | 4.72M   | 82.28  | 87.06      | 91.60    | 55.65 | 24.02 | 68.53 | 68.19  |
> | PiSSA($r=8$)      | 4.72M   | 82.78  | 88.83      | 91.14    | 54.97 | 25.56 | 67.90 | 68.53  |
> | BoRA($r=8,b=16$)  | 4.92M   | 88.35  | 93.00      | 92.72    | 58.83 | 27.17 | 73.40 | 72.24  |
> | BoRA+($r=8,b=16$) | **0.20M**   | 84.56  | 88.67      | 90.94    | 53.48 | 25.77 | 67.10 | **68.42**  |
> | BoRA+($r=8,b=64$) | 3.15M   | 87.35  | 90.17      | 92.52    | 56.18 | 24.62 | 71.50 | 70.39  |
>
> We will include a detailed explanation and the complete experimental results of BoRA+ in the appendix.
>
> ***`Q5: "will the matrix rank change with different layers and training periods"`***
>
> **R5:**
> Yes, the matrix rank (i.e., $\text{rank}(\Delta W)$) **can change over time** for both LoRA and BoRA. However, the **rank differences across layers are minimal**.
>
> At initialization, the matrix $B$ in both LoRA and BoRA is set to zero, so the rank of both $B$ and $\Delta W$ is initially zero. As training progresses, the matrix $B$ is updated, causing the rank to gradually increase. Statistical analysis shows that after approximately 50 training steps, $\text{rank}(\Delta W)$ quickly reaches its upper bound, which is $r$ for LoRA and $br$ for BoRA. Once this upper bound is reached, the rank stays mostly the same for the rest of the training, with only minimal variation across layers.
>
> However, as shown in Figure 4 of the paper, **the magnitudes of the singular values do vary between layers**, even though the overall rank remains stable.
>
>
> **`Lastly, thank you again for your valuable feedback, which has greatly improved our work. We hope our responses address your concerns and clarify any doubts. Please feel free to reach out to us if you have any further questions.`**

---

> > ### Comment · Reviewer_dURX · 2025-11-28
> > **Thank you for your reply.**
> >
> > I appreciate the thorough answers you have provided to my questions. I'm satisfied with the your response.

---

> > > ### Author Response · Authors · 2025-11-28
> > > **Sincere Thanks for Your Support**
> > >
> > > Thank you again for your valuable suggestions; we are glad we could address your concerns.

---

### Author Response · Authors · 2025-11-30
**Summary of Review and Discussion**

Dear **Area Chairs** and **Reviewers**,

Thank you for the time and effort you've dedicated to reviewing our work.

Here, we'd like to briefly summarize **the pre-revert discussion**, **the contributions of this paper**, and **the reviewers' evaluations**.

### **`I Pre-Revert Discussion`**
The reviewers' ratings and confidence levels, before and after the discussion, are shown below, with **any changes highlighted in bold**.
||Before Discussion|After Disscussion|
|-|-|-|
|Reviewer dURX|6 (confidence: 4)|6 (confidence: 4)|
|Reviewer YooA|6 (confidence: 4)|6 (confidence: 4)|
|Reviewer 9WMS|4 (confidence: 3)|**6** (confidence: **4**)|
|Reviewer rMns|4 (confidence: 3)|**6** (confidence: 3)|

Following the timely and productive discussion before **Nov. 23**, our paper received **unanimous positive ratings from all reviewers (6–6–6–6)**, with an **average confidence level of 3.75**.

Our rebuttal was submitted on Nov. 21. Soon after, on Nov. 22 and Nov. 23, we received responses from Reviewers **9WMS** and **rMns**, both of whom confirmed that **all their concerns had been fully addressed**. They **each raised their rating to 6**, with Reviewer 9WMS also increasing the confidence level to 4.
Subsequently, we received feedback from Reviewers **YooA** and **dURX**, both of whom **had already provided positive ratings** and relatively high confidence from the start. Reviewer dURX confirmed that all issues had been resolved, while Reviewer YooA indicated that most concerns had been addressed and then proceeded to further probe two of the original questions. In response, we provided further clarification and supported our explanations with additional experiments.

### **`II Contributions of This Paper`**

The key idea of this work is to analyze the rank limitations of LoRA weights (i.e., $BA$)through the lens of block matrix multiplication, where $B \in \mathbb{R}^{m \times r} = [B_1, B_2, \dots, B_b]^\top$ and $A \in \mathbb{R}^{r \times n} = [A_1, A_2, \dots, A_b]$. Building on our analysis, we propose **BoRA**, which introduces a diagonal matrix ($\Sigma_{i,j}$) into each block matrix multiplication. This addition **enhances block diversity** and, in turn, **increases the rank of the resulting LoRA weights**.

A simple illustration of the resulting weights in LoRA and BoRA is provided below.
$$
\text{LoRA:} \quad
\left[
\begin{array}{ccccc}
B_1A_1&B_1A_2&\dots&B_1A_b \\\\
B_2A_1&B_2A_2&\dots&B_2A_b \\\\
\vdots&\vdots&\vdots&\vdots \\\\
B_bA_1&B_bA_2&\dots&B_bA_b \\\\
\end{array}
\right]
\quad \longrightarrow \quad
\text{BoRA:} \quad
\left[
\begin{array}{ccccc}
B_1\Sigma_{1,1}A_1&B_1\Sigma_{1,2}A_2&\dots&B_1\Sigma_{1,b}A_b\\\\
B_2\Sigma_{2,1}A_1&B_2\Sigma_{2,2}A_2&\dots&B_2\Sigma_{2,b}A_b\\\\
\vdots&\vdots&\vdots&\vdots\\\\
B_b\Sigma_{b,1}A_1&B_b\Sigma_{b,2}A_2&\dots&B_b\Sigma_{b,b}A_b\\\\
\end{array}
\right]
$$

The advantages of BoRA are as follows:
* **Theoretical Basis**: BoRA is theoretically proven to increase the rank upper bound of LoRA weights by a factor of $b$.
* **Low Overhead**: BoRA's diagonal matrices usually introduce minimal computational cost and parameter overhead.
* **Unified Framework**: BoRA generalizes LoRA and variants like MELoRA, reducing to them when the diagonal matrices assume specific values.
* **Strong Performance**: BoRA achieves much better accuracy than LoRA and other baselines, while using a similar number of trainable parameters.

### **`III Reviewers' Evaluations`**
Lastly, we summarize the key **strengths** and **weaknesses** identified by the reviewers.

**Strengths:**
1. Novel Idea **[9WMS, rMns]**
2. Clear Presentation **[dURX, rMns]**
3. Strong and Unified Theoretical Framework Established by BoRA **[YooA, 9WMS]**
4. Comprehensive Experiments and Ablation Studies **[9WMS, dURX, YooA]**
5. Superior Performance Demonstrated by BoRA **[dURX, YooA, 9WMS, rMns]**

**Weaknesses:**
1. Lack of Comparison with Additional Baselines **[dURX, YooA, rMns]**
2. Lack of Comparison for End-to-End Training Time **[YooA, rMns]**
3. Small Number of Training Epochs for GLUE Tasks **[YooA]**
4. Instability of BoRA's Performance in Fig. 3(b) **[YooA]**
5. Lack of Certain Assumptions for BoRA to Achieve the Rank Upper Bound **[9WMS]**

Please note that most of the weaknesses are related to additional experiments, and we have addressed all of them with further experimental results and analysis.
For **weakness 1**, we have included comparisons with several additional methods, including HiRA, KronA, and ReLoRA.
For **weakness 2**, we have reported memory usage and training time across various methods.
For **weakness 3**, we have extended the experiments with additional training epochs.
For **weakness 4**, we have explained the accuracy instability in Fig. 3(b) and provided averaged results from multiple experiments, showing no significant instability.
For **weakness 5**, we have clarified the conditions under which BoRA can achieve the rank upper bound.

---

### Meta-Review · Area_Chair_g226 · 2026-01-07

**Summary:**

This paper presents a novel approach to low-rank adaptation that is both theoretically grounded and empirically effective. The reviewers initially recognized the novelty and potential of the proposed BoRA method but raised several valid concerns regarding its evaluation and theoretical presentation:

**Experiment:** Multiple reviewers (Reviewer dURX, Reviewer YooA) questioned the fairness of the comparisons.

**Efficiency Analysis:** A common thread across reviews (including Reviewer rMns and dURX) was the lack of direct "wall-clock" time and memory usage comparisons to verify the efficiency claims against standard LoRA and other variants.

**Baselines:** Reviewer rMns and others requested comparisons against a broader range of baselines (such as HiRA, KronA, and ReLoRA) to firmly establish state-of-the-art performance.

**Theoretical and Notation Issues:** Reviewer 9WMS identified inconsistencies in mathematical notation (specifically regarding index ranges for block matrices) and questioned the assumptions behind the rank upper bound proofs.

**Reviewer Concerns:**

The authors have resolved the initial concerns regarding baselines, training rigor, and efficiency benchmarks. There are no significant outstanding concerns. The reviewers have indicated that the revised manuscript and additional data have sufficiently supported the paper's claims.

**Reviewer Scores:**

The reviewers did engage in the discussion, and their score trajectories reflect the positive resolution of their concerns. Reviewers 9WMS and rMns would likely move from 4 (Borderline Reject) to 6 (Borderline Accept), and Reviewers YooA and dURX would likely maintain their scores.

---

### Decision · Program_Chairs · 2026-01-26

Accept (Poster)